# ADAPTSR: RANK-AWARE LOW-RANK ADAPTATION FOR REAL-WORLD SUPER-RESOLUTION

## ABSTRACT

Recovering high-frequency details from low-resolution images remains a central challenge in super-resolution (SR), particularly under complex and unknown real-world degradations. While GAN-based methods improve perceptual sharpness, they are unstable and introduce artifacts, and diffusion models achieve strong fidelity but demand excessive computation, even in few-step variants. We present AdaptSR, a rank-aware low-rank adaptation framework that efficiently repurposes bicubic-trained CNN and Transformer SR backbones for real-world tasks. Unlike full fine-tuning, AdaptSR inserts lightweight LoRA modules into convolution, attention, and MLP layers, updates them under a rank-aware allocation strategy guided by layer importance, and merges them back after training—ensuring no additional inference cost. This design reduces trainable parameters by up to 92% and shortens adaptation time from days to just 1–4 hours on a single GPU, aligning with the goals of sustainable and budget-friendly AI. Extensive experiments across diverse SR backbones and datasets show that AdaptSR consistently matches or surpasses full fine-tuning, outperforms recent GAN- and diffusion-based methods in distortion metrics, and delivers competitive perceptual quality. Comparisons with other parameter-efficient fine-tuning (PEFT) baselines further confirm the advantages of our rank-aware allocation. By unifying efficiency, scalability, and practical deployment, AdaptSR establishes a sustainable path for adapting SR models to real-world degradations. The code will be made publicly available.

## 1 INTRODUCTION

Image super-resolution (SR) aims to reconstruct high-resolution (HR) images from low-resolution (LR) inputs and has essential applications in fields like medical Isaac & Kulkarni (2015); Qiu et al. (2023), optical Lin et al. (2024); Rosen et al. (2024), and satellite imaging Chen et al. (2023a); Karwowska & Wierzbicki (2022). While popular convolutional neural network (CNN) and Transformer-based SR models Lim et al. (2017); Zhang et al. (2018b); Wang et al. (2018); Zhang et al. (2018c); Niu et al. (2020); Liang et al. (2021); Chen et al. (2022b); Zhang et al. (2022); Chen et al. (2023c;b); Korkmaz & Tekalp (2024) have advanced architectures, they are usually trained on synthetic bicubic downsampling, limiting their generalization to real-world degradations. Fine-tuning on real-world data helps bridge this gap Wang et al. (2018); Liang et al. (2022a); Wang et al. (2024a); Yu et al. (2024), but requires long training schedules, high memory, and large compute budgets, making it impractical for deployment on lightweight or resource-constrained devices.

Real-world SR (Real SR) Cai et al. (2019); Wei et al. (2020) has therefore gained traction as an alternative to synthetic SR, explicitly training on complex degradations Zhang et al. (2021); Wang et al. (2021); Wu et al. (2024a;b); Assaf Shocher (2018). GAN-based approaches Goodfellow et al. (2014); Wang et al. (2018; 2021) leverage adversarial losses to improve perceptual realism but suffer from instability and hallucinated textures. Diffusion models (DMs) Kawar et al. (2022); Ho et al. (2020); Dhariwal & Nichol (2021); Rombach et al. (2022); Yang et al. (2024); Wang et al. (2024a) achieve state-of-the-art fidelity by exploiting powerful priors, yet their iterative sampling and massive backbones are computationally prohibitive. Even recent one-step DMs Wu et al. (2024a;b) trade efficiency for detail quality, leaving an unsolved gap between practicality and performance. This motivates two research questions: (1) Can bicubic-trained CNN/Transformer SR models be efficiently repurposed for real-world degradations without full fine-tuning? (2) Can such adapta-

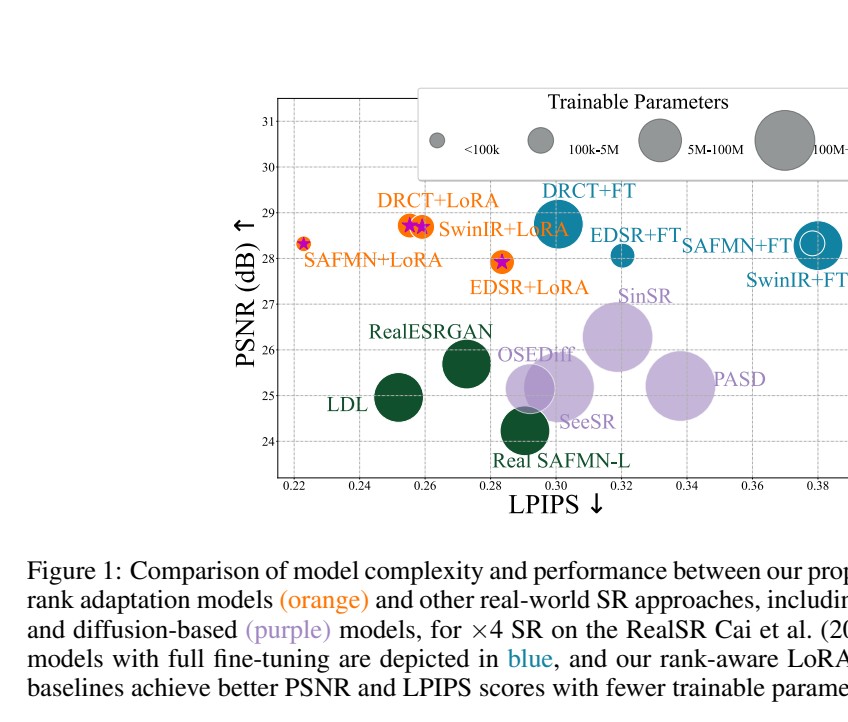

Figure 1: Comparison of model complexity and performance between our proposed rank-aware low-rank adaptation models (orange) and other real-world SR approaches, including GAN-based (green) and diffusion-based (purple) models, for ×4 SR on the RealSR Cai et al. (2019) dataset. Baseline models with full fine-tuning are depicted in blue, and our rank-aware LoRA models for the same baselines achieve better PSNR and LPIPS scores with fewer trainable parameters.

tion not only match but outperform GAN and diffusion Real-SR methods in fidelity and perceptual quality, while remaining lightweight and sustainable to train?

We address these questions with AdaptSR, a parameter-efficient domain adaptation framework based on Low-Rank Adaptation (LoRA) Hu et al. (2021). Unlike standard fine-tuning, AdaptSR injects lightweight low-rank adapters into convolution, attention, and MLP layers, and trains only these modules while freezing the backbone. Crucially, we introduce a rank-aware allocation strategy that scores layers by gradient importance and adaptively assigns ranks under a global parameter budget. After training, LoRA weights are merged back into the frozen backbone, guaranteeing identical inference FLOPs. Our goal is efficient fidelity-oriented adaptation, not to replace full fine-tuning, but to provide a lightweight alternative when structural fidelity and fast real-world transfer are prioritized. This approach reduces training cost by up to 92%, achieves adaptation in just 1–4 GPU hours, and enables deployment on edge devices—aligning with the principles of sustainable and budget-friendly AI.

Beyond practical efficiency, we provide the first systematic analysis of rank-aware allocation in SR, comparing it to uniform ranks and other PEFT methods such as ARC and DiffFit. Extensive experiments across diverse backbones (EDSR Lim et al. (2017), SAFMN Sun et al. (2023), SwinIR Liang et al. (2021), DRCT Hsu et al. (2024)) and benchmarks (RealSR Cai et al. (2019), DRealSR Wei et al. (2020), DSLR Ignatov et al. (2017)) demonstrate that AdaptSR consistently matches or exceeds full fine-tuning, while outperforming GAN and diffusion-based Real-SR pipelines in distortion metrics and delivering competitive perceptual quality. Our analysis further reveals SR-specific behavior not captured in prior LoRA or PEFT studies, including the highly structured distribution of gradient saliency across SR blocks, the unusually high effectiveness of very low ranks (e.g., $r=1$ recovering ∼95% of full-finetuning PSNR), and the clear benefit of spatially non-uniform adapter placement for bicubic-to-real adaptation. To summarize, our main contributions are:

- We present AdaptSR, a backbone-agnostic LoRA framework that adapts bicubic-trained CNN and Transformer SR models to real-world degradations in just 1–4 GPU hours, while guaranteeing no inference overhead after merging.

- We introduce a rank-aware allocation scheme that scores layers by gradient importance and allocates LoRA capacity under a global parameter budget. This strategy consistently outperforms uniform rank assignment and reveals where SR models gain the most from low-rank updates.

- AdaptSR achieves up to 92% parameter reduction and days-to-hours training speedups, enabling sustainable and budget-friendly deployment. Experiments across diverse SR back-

bones (EDSR, SAFMN, SwinIR, DRCT) and datasets (RealSR, DRealSR, DSLR) show +3–4 dB PSNR and +2% perceptual gains over GAN/diffusion baselines, while surpassing recent PEFT approaches (e.g., ARC, DiffFit).

## 2 RELATED WORK

### 2.1 REAL-WORLD IMAGE SUPER-RESOLUTION

Starting with SRCNN Lin et al. (2017), deep learning-based convolutional SR networks Lim et al. (2017); Zhang et al. (2018b); Wang et al. (2018); Zhang et al. (2018c); Liang et al. (2022b); Niu et al. (2020) achieved notable progress. Later, after the introduction of SwinIR Liang et al. (2021), transformer-based SR networks Chen et al. (2022b); Zhang et al. (2022); Chen et al. (2023c;b); Korkmaz & Tekalp (2024) gain popularity by significantly boosting the visual quality of the reconstructed images. However, these models typically assume simple, known degradations (e.g., bicubic downsampling), limiting their performance on real-world images with complex, unknown degradations. To address real SR challenges, GAN-based models Goodfellow et al. (2014) like SRGAN Ledig et al. (2017), BSRGAN Zhang et al. (2021) and Real-ESRGAN Wang et al. (2021) introduced sophisticated degradation models for real-world data. More recent methods like LDL Liang et al. (2022a) and DeSRA Xie et al. (2023b) improve artifact suppression using local statistics, however, GANs still struggle with instability and introduce unnatural artifacts. Recently introduced diffusion models Ho et al. (2020); Dhariwal & Nichol (2021); Kawar et al. (2022); Yue et al. (2023); Yu et al. (2024); Wang et al. (2024b); Lin et al. (2025) have also been applied to real SR tasks, such as StableSR Wang et al. (2024a) and PASD Yang et al. (2024) use feature warping to enhance quality. However, they require hundreds of steps to complete the diffusion process along with excessive memory consumption. Unlike these methods, our approach achieves high-quality SR in real-world settings with minimal training time, enabling rapid adaptation to real SR tasks in a couple of hours.

### 2.2 PARAMETER EFFICIENT FINE-TUNING FOR REAL SR

Parameter-efficient fine-tuning (PEFT) Hu et al. (2021; 2023); Dettmers et al. (2024); Xie et al. (2023a) methods like adapters Hu et al. (2023); Lei et al. (2023); Dong et al. (2024); He et al. (2025); Zhou et al. (2025); Bhardwaj et al. (2024) and LoRA Hu et al. (2021); Dettmers et al. (2024) have become crucial for fine-tuning large pre-trained models with fewer resources. Adapters add small modules between layers for task-specific updates, while LoRA performs low-rank adaptations of model weight matrices, enhancing efficiency without increasing inference-time parameters. LoRA has shown versatility across NLP Hou et al. (2023); Dettmers et al. (2024), computer vision Wu et al. (2024b); Park et al. (2024); Tian et al. (2024); Agiza et al. (2024); Borse et al. (2025) and image reconstruction applications Park et al. (2024); Tian et al. (2024) due to its adaptability and computational benefits.

**LoRA-Based Adaptation in Super-Resolution.** In SR, LoRA has been widely adopted in diffusion-based pipelines to leverage pre-trained text-to-image generative priors. SeeSR Wu et al. (2024b) enriches perceptual details through prompt-driven semantic conditioning, whereas OSEDiff Wu et al. (2024a) distills a text-to-image model into a one-step SR process, significantly reducing the number of diffusion iterations. Although both approaches employ LoRA, their overall computational cost remains dominated by large diffusion backbones: SeeSR fine-tunes a 750M-parameter prompt extractor over the course of a week, while OSEDiff (8.5M trainable parameters) still inherits the heavy Stable Diffusion Wang et al. (2024a) machinery and exhibits notable degradation in PSNR when confronted with complex real-world degradations. A recent line of work, PiSA-SR Sun et al. (2025), further extends the use of LoRA within diffusion-based SR by decoupling the training objective into two distinct LoRA branches: a pixel-level LoRA supervised by an $\ell_2$ regression loss, and a semantic-level LoRA trained with LPIPS and classifier-based score distillation. During inference, PiSA-SR introduces adjustable guidance coefficients that allow users to continuously tune the trade-off between pixel-wise fidelity and perceptual sharpness without re-training. Despite its flexibility and strong perceptual quality, PiSA-SR remains inherently tied to diffusion inference, and its efficiency is thus constrained by the cost of operating Stable Diffusion, even in its single-step variant. Whereas PiSA-SR targets user-controllable perceptual–fidelity trade-offs within a diffusion prior, AdaptSR targets memory-efficient, degradation-robust backbone adaptation in a fully feed-forward SR setting. The two approaches are thus complementary in motivation, technical design, and com-

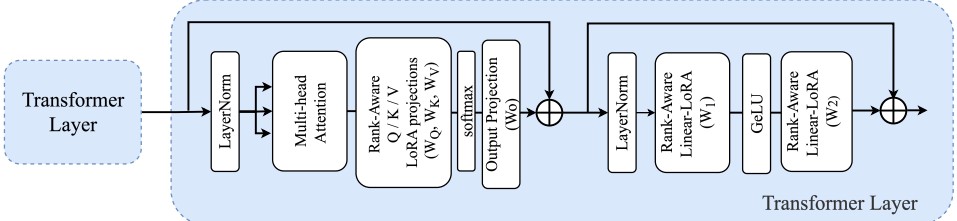

Figure 2: Transformer LoRA Layers (TLL) include rank-aware LoRA-modified multi-head self-attention (MSA) and multi-layer perceptron (MLP) layers, using Linear-LoRA for efficient domain adaptation with reduced parameters. Each LoRA site (Q/K/V, output, MLP1/2) receives a rank from the allocation map; sites with r=0 are pruned at training time and all adapters are merged post-training, preserving the baseline inference cost.

putational regime. Unlike SeeSR, OSEDiff, and PiSA-SR, our method does *not* rely on diffusion priors, semantic score distillation, or prompt guidance. Instead, AdaptSR focuses on the complementary and underexplored problem of efficiently adapting *bicubic-trained CNN/Transformer SR backbones* to real-world degradations. Our approach introduces a rank-aware allocation strategy that selectively updates LoRA layers according to backbone-specific importance signals, enabling high-quality real SR adaptation with only 886k trainable parameters and no inference-time overhead.

## 3 METHODOLOGY

### 3.1 PRELIMINARY: LOW-RANK ADAPTATION (LORA)

Low-Rank Adaptation (LoRA) Hu et al. (2021) fine-tunes a pre-trained model by introducing a low-rank decomposition of weight updates. Instead of updating a full weight matrix $W_0 \in \mathbb{R}^{d \times k}$, LoRA parameterizes the update as:

$$\Delta W = \frac{\alpha}{r} BA, \quad W' = W_0 + \Delta W, \tag{1}$$

where $A \in \mathbb{R}^{r \times k}$ and $B \in \mathbb{R}^{d \times r}$ are learnable low-rank matrices, $r \ll \min(d, k)$ is the rank, and $\alpha$ is a scaling factor. During training, only $A$ and $B$ are updated while $W_0$ is frozen, resulting in efficient adaptation with drastically fewer trainable parameters. After training, the low-rank update is merged back into $W_0$, guaranteeing identical FLOPs and inference latency. This property makes LoRA well aligned with the goals of sustainable and budget-friendly training, as it reduces both memory usage and carbon footprint compared to full fine-tuning.

### 3.2 LAYER-WISE LORA INTEGRATION

**Conv-LoRA Layers.** Convolutional layers are central to both CNN and Transformer-based SR models, responsible for texture extraction, upsampling, and reconstruction. For a convolution with weights $W \in \mathbb{R}^{C_{\text{out}} \times C_{\text{in}} \times k \times k}$, AdaptSR applies a LoRA update via two sequential low-rank convolutions:

$$\text{Conv}_{\text{LoRA}}(x) = (W * x) + \frac{\alpha}{r} B_{\text{conv}} * (A_{\text{conv}} * x), \tag{2}$$

where $A_{\text{conv}} \in \mathbb{R}^{r \times C_{\text{in}} \times 1 \times 1}$ reduces channel dimension, and $B_{\text{conv}} \in \mathbb{R}^{C_{\text{out}} \times r \times k \times k}$ (k=3) restores it, ensuring dimensional consistency. In practice, we adopt a $1 \times 1 \to k \times k$ LoRA factorization, which empirically provides the most effective low-rank basis for SR filters compared to $k \times k \to 1 \times 1$ or single-kernel variants in Table 9. This decomposition avoids the undefined 4D multiplication noted in prior reviews, while maintaining full compatibility with convolutional kernels.

**MLP-LoRA Layers.** In SR architectures, MLP layers refine features captured by attention and convolution. Given linear projections $W_1 \in \mathbb{R}^{d \times d}$ and $W_2 \in \mathbb{R}^{d \times d}$, AdaptSR modifies them as:

$$W_1' = W_1 + \frac{\alpha}{r} B_1 A_1, \quad W_2' = W_2 + \frac{\alpha}{r} B_2 A_2, \tag{3}$$

where $A_1, A_2 \in \mathbb{R}^{r \times d}$ and $B_1, B_2 \in \mathbb{R}^{d \times r}$. The resulting MLP block becomes:

$$\text{MLP}_{\text{LoRA}}(x) = W_2' \, \sigma(W_1' x), \tag{4}$$

with $\sigma$ denoting GELU Hendrycks & Gimpel (2016). This preserves long-range dependencies while adapting efficiently under a tight parameter budget.

**MSA-LoRA Layers.** Multi-head self-attention (MSA) layers capture global structure in Transformer-based SR models. For query, key, and value projections $W_Q, W_K, W_V \in \mathbb{R}^{d \times d_k}$, AdaptSR introduces LoRA-modified projections:

$$Q = x(W_Q + \Delta W_Q), \quad K = x(W_K + \Delta W_K), \quad V = x(W_V + \Delta W_V), \tag{5}$$

where $\Delta W_Q = \frac{\alpha}{r} B_Q A_Q$, $\Delta W_K = \frac{\alpha}{r} B_K A_K$, $\Delta W_V = \frac{\alpha}{r} B_V A_V$, with $A_* \in \mathbb{R}^{r \times d}$, $B_* \in \mathbb{R}^{d_k \times r}$, and $d_k = d/h$ for $h$ heads. This ensures $\Delta W_* \in \mathbb{R}^{d \times d_k}$, resolving the dimensionality mismatch concerns raised in prior reviews. The final attention output is:

$$\text{Attention}_{\text{LoRA}}(Q, K, V) = \text{softmax}\left(\frac{QK^T}{\sqrt{d_k}} + B\right) V. \tag{6}$$

## 3.3 ADAPTSR

AdaptSR integrates Conv-, MLP-, and MSA-LoRA layers into bicubic-trained SR backbones (e.g., EDSR Lim et al. (2017), SAFMN Sun et al. (2023), SwinIR Liang et al. (2021), DRCT Hsu et al. (2024)). During training, only the low-rank adapters are updated while the backbone remains frozen, drastically reducing compute and memory. After training, all LoRA weights are merged back, so inference cost remains identical to the original model. This property enables rapid, sustainable domain adaptation on resource-constrained hardware: e.g., $\leq 1$ hour on a single RTX 4090 for SAFMN, and $\approx 4$ hours for SwinIR.

**Rank-Aware LoRA-Inserted Transformer Layer.** Given a rank map $\{r_\ell\}$ from Alg. 1, we attach independent LoRA adapters to the self-attention projections $(W_Q, W_K, W_V, W_O)$ and the MLP projections $(W_1, W_2)$ with ranks $\{r_Q, r_K, r_V, r_O, r_1, r_2\}$ respectively as depicted in Figure 2. Each adapter uses the same scaling convention $\Delta W = \frac{\alpha}{r} BA$ and is *skipped* when its rank is zero. The attention projections are:

$$Q = X\left(W_Q + \mathbf{1}[r_Q > 0] \frac{\alpha_Q}{r_Q} B_Q A_Q\right), \quad K = X\left(W_K + \mathbf{1}[r_K > 0] \frac{\alpha_K}{r_K} B_K A_K\right),$$

$$V = X\left(W_V + \mathbf{1}[r_V > 0] \frac{\alpha_V}{r_V} B_V A_V\right).$$

with $A_\star \in \mathbb{R}^{r_\star \times d}$, $B_\star \in \mathbb{R}^{d_k \times r_\star}$ and $d_k = d/h$. The output projection uses

$$O = \text{Softmax}\left(\frac{QK^\top}{\sqrt{d_k}} + B\right) V, \quad Y = O\left(W_O + \mathbf{1}[r_O > 0] \frac{\alpha_O}{r_O} B_O A_O\right).$$

For the MLP, we define

$$W_1' = W_1 + \mathbf{1}[r_1 > 0] \frac{\alpha_1}{r_1} B_1 A_1, \quad W_2' = W_2 + \mathbf{1}[r_2 > 0] \frac{\alpha_2}{r_2} B_2 A_2,$$

and $\text{MLP}(X) = W_2' \sigma(W_1' X)$. After training, all $\Delta W$ terms are merged into their backbones, preserving the inference-time FLOPs and parameter count.

**Theoretical Motivation.** Our rank allocation uses the gradient norm $\|\nabla_{W_\ell} \mathcal{L}\|_2$ as an importance score. This choice follows directly from a first-order Taylor expansion of the loss:

$$\Delta \mathcal{L} \approx |\langle \nabla_{W_\ell} \mathcal{L}, \Delta W_\ell \rangle| \leq \|\nabla_{W_\ell} \mathcal{L}\|_2 \|\Delta W_\ell\|_2. \tag{7}$$

Thus, for a fixed update budget, allocating higher LoRA rank to layers with larger $\|\nabla_{W_\ell} \mathcal{L}\|_2$ maximizes expected loss reduction. This provides a direct first-order justification for gradient-based saliency.This metric is closely related to the diagonal Fisher information: $F_\ell \approx \mathbb{E}[(\nabla_{W_\ell} \mathcal{L})^2]$, making it a lightweight proxy for second-order sensitivity without computing Hessians. It also aligns with sensitivity measures used in pruning methods (e.g., SNIP Lee et al. (2019), GraSP Wang et al. (2020)), where gradient magnitude reflects the impact of removing or perturbing weights. In SR, importance scores are stable due to strong spatial redundancy: across 50 random mini-batches, the Spearman correlation of layer rankings remains $\rho = 0.91$. Besides, unlike activation amplitude, which reflects forward signal strength rather than loss sensitivity, gradient-based scoring directly

captures the influence of each layer on the objective. Hence, computing scores once at initialization is sufficient and adds negligible overhead. We compare gradient-based allocation with Fisher diagonal, activation norms, and random scoring in Table 11. Under equal parameter budgets, gradient-based allocation consistently yields the best PSNR and LPIPS, confirming both theoretical soundness and empirical effectiveness.

---

**Algorithm 1** Rank-Aware Layer Selection and Allocation

---

**Require:** Pre-trained SR model $\mathcal{M}$ with layers $\{W_\ell\}_{\ell=1}^L$, parameter budget $P$, min/max rank $r_{\min}, r_{\max}$.

**Ensure:** Rank map $\{r_\ell\}$ such that $\sum_\ell r_\ell \leq P$.

1: **Importance scoring**: sample mini-batch $\mathcal{B}$; compute $s_\ell = \|\nabla_{W_\ell}\mathcal{L}(\mathcal{B})\|_2$.
2: **Sorting**: order layers by $s_\ell$ in descending order.
3: **Budget allocation**: assign higher $r_\ell$ to top-ranked layers within $[r_{\min}, r_{\max}]$ until budget $P$ is met.
4: **return** rank map $\{r_\ell\}$.

---

## 4 EXPERIMENTS

### 4.1 EXPERIMENTAL SETTINGS

**Datasets.** We train AdaptSR on DIV2K Agustsson & Timofte (2017) and RealSR Cai et al. (2019) training sets, using the Real-ESRGAN Wang et al. (2021) degradation pipeline to create LR-HR pairs. During training, we also utilize the DIV2K unknown degradation dataset to cover a wider range of degradation types, using randomly cropped $256 \times 256$ patches from the HR images. For evaluation, we benchmark against other methods on DIV2K Agustsson & Timofte (2017), RealSR Cai et al. (2019), and DRealSR Wei et al. (2020) test sets. Especially to compare with diffusion-based approaches, we adopt the StableSR Wang et al. (2024a) validation setup, including 3000 patches from DIV2K, 100 from RealSR, and 93 from DRealSR, with LR-HR pairs sized at $128 \times 128$ and $512 \times 512$.

**Implementation Details and Evaluation.** All experiments were run on an NVIDIA Quadro RTX 4090 GPU using PyTorch. LoRA models rank 8 with $\alpha$ value 1 were trained for 100k iterations with Adam Kingma (2014) and L1 loss, starting with a learning rate of $1e^{-3}$, reduced by 25% at 50k, 75k, and 90k iterations. In comparison, full model fine-tuning required 500k iterations to achieve comparable results, with the learning rate halved at 250k, 400k, 450k and 475k intervals. Both LoRA and fine-tuning used the same settings, including the Adam optimizer Kingma (2014), a batch size of 8, and the L1 loss function. To evaluate fidelity and perceptual quality, we used PSNR and SSIM (Y channel in YCbCr space) and perceptual metrics LPIPS Zhang et al. (2018a) and DISTS Ding et al. (2020).

### 4.2 COMPARISON WITH STATE-OF-THE-ART

**Quantitative Performance.** Table 1 presents a quantitative comparison between our AdaptSR method and state-of-the-art real SR models, including GAN-based approaches (*e.g.*, RealESRGAN Wang et al. (2021), LDL Liang et al. (2022a)) and diffusion methods (*e.g.*, StableSR Yang et al. (2024), DiffBIR Lin et al. (2025), ResShift Yue et al. (2023), SeeSR Wu et al. (2024b), PASD Yang et al. (2024), SinSR Wang et al. (2024b), OSEDiff Wu et al. (2024a)). The results are evaluated on ×4 real SR benchmarks. Despite using 20× fewer parameters than GAN-based methods like RealESRGAN Wang et al. (2021) and LDL Liang et al. (2022a), AdaptSR (e.g., SwinIR+LoRA) improves PSNR by up to 4 dB and LPIPS by 3% on RealSR Cai et al. (2019) while updating only 886k parameters. Similarly, while diffusion models such as SeeSR Wu et al. (2024b) and PASD Yang et al. (2024) require over 600M parameters, and OSEDiff Wu et al. (2024a)—which is still 10× larger than our model—struggles to match our performance, AdaptSR achieves 2 dB higher PSNR on DIV2K Agustsson & Timofte (2017) and surpasses OSEDiff by approximately 3.5 dB and 2.8 dB on the RealSR Cai et al. (2019) and DRealSR Wei et al. (2020) datasets, respectively. To ensure fair comparison, we distinguish fidelity-oriented adaptation (our objective) from generative perceptual enhancement targeted by GAN and diffusion models: the latter intentionally hal-

Table 1: Quantitative comparison of perception-oriented AdaptSR-GAN against GAN- and diffusion-based real-SR methods, and fidelity-oriented AdaptSR against conventional CNN and Transformer SR backbones trained with L1 and realSR data. Rows are methods; columns are metrics. The best and the second-best results are highlighted in red and blue, respectively.

| | Method | Param. | DIV2K | | | | RealSR | | | | DRealSR | | | |
|---|---|---|---|---|---|---|---|---|---|---|---|---|---|---|
| | | | PSNR ↑ | SSIM ↑ | LPIPS ↓ | DISTS ↓ | PSNR ↑ | SSIM ↑ | LPIPS ↓ | DISTS ↓ | PSNR ↑ | SSIM ↑ | LPIPS ↓ | DISTS ↓ |
| Perception Oriented | RealESRGAN | 17M | 22.94 | 0.6036 | 0.3768 | 0.2520 | 25.69 | 0.7616 | 0.2727 | 0.2063 | 28.64 | 0.8053 | 0.2847 | 0.2089 |
| | LDL | 12M | 23.76 | 0.6403 | 0.3091 | 0.2189 | 24.96 | 0.7634 | 0.2519 | 0.1981 | 27.43 | 0.8078 | 0.2655 | 0.2055 |
| | DASR | 8M | 21.72 | 0.5535 | 0.4266 | 0.2688 | 27.02 | 0.7708 | 0.3151 | 0.2207 | 29.77 | 0.7572 | 0.3169 | 0.2235 |
| | Real-SAFMN-L | 5.6M | 23.21 | 0.5950 | 0.3282 | 0.2153 | 24.23 | 0.7217 | 0.2905 | 0.2176 | 27.15 | 0.7671 | 0.3148 | 0.2219 |
| | StableSR | 150M | 23.68 | 0.4887 | 0.4055 | 0.2542 | 24.70 | 0.7085 | 0.3018 | 0.2135 | 28.13 | 0.7542 | 0.3315 | 0.2263 |
| | DiffBIR | 380M | 20.84 | 0.4938 | 0.4270 | 0.2471 | 24.77 | 0.6572 | 0.3658 | 0.2310 | 26.76 | 0.6576 | 0.4599 | 0.2749 |
| | ResShift | 119M | 20.94 | 0.5422 | 0.4284 | 0.2606 | 26.31 | 0.7421 | 0.3460 | 0.2498 | 28.46 | 0.7673 | 0.4006 | 0.2656 |
| | SeeSR | 750M | 21.75 | 0.6043 | 0.3194 | 0.1968 | 25.18 | 0.7216 | 0.3009 | 0.2223 | 28.17 | 0.7691 | 0.3189 | 0.2315 |
| | PASD | 625M | 23.14 | 0.5505 | 0.3571 | 0.2207 | 25.21 | 0.6798 | 0.3380 | 0.2260 | 27.36 | 0.7073 | 0.3760 | 0.2531 |
| | SinSR | 119M | 24.41 | 0.6018 | 0.3240 | 0.2066 | 26.28 | 0.7347 | 0.3188 | 0.2353 | 28.36 | 0.7515 | 0.3665 | 0.2485 |
| | OSEDiff | 8.5M | 23.72 | 0.6108 | 0.2941 | 0.1976 | 25.15 | 0.7341 | 0.2921 | 0.2128 | 27.92 | 0.7835 | 0.2968 | 0.2165 |
| | PiSA-SR | 6.2M | 23.87 | 0.6058 | 0.2823 | 0.1934 | 25.50 | 0.7417 | 0.2672 | 0.2044 | 28.31 | 0.7804 | 0.2960 | 0.2169 |
| | AdaptSR-GAN (Ours) | 886k | 24.11 | 0.6598 | 0.2914 | 0.2185 | 26.48 | 0.7672 | 0.2256 | 0.2016 | 28.75 | 0.7856 | 0.2788 | 0.2079 |
| Distortion Oriented | EDSR | 1.5M | 24.92 | 0.6661 | 0.5140 | 0.3204 | 28.06 | 0.7963 | 0.3203 | 0.2303 | 30.65 | 0.8328 | 0.3464 | 0.2695 |
| | SAFMN | 240k | 24.93 | 0.6664 | 0.5158 | 0.3208 | 28.32 | 0.7936 | 0.2887 | 0.2472 | 30.62 | 0.8172 | 0.3456 | 0.2689 |
| | SwinIR | 12M | 24.87 | 0.6637 | 0.5131 | 0.3205 | 28.25 | 0.7906 | 0.3820 | 0.2304 | 30.62 | 0.8176 | 0.3491 | 0.2708 |
| | DRCT | 14M | 24.98 | 0.6679 | 0.5144 | 0.3201 | 28.75 | 0.8100 | 0.3007 | 0.2584 | 30.61 | 0.8189 | 0.3494 | 0.2703 |
| | AdaptSR (Ours) | 886k | 25.81 | 0.6681 | 0.5047 | 0.3194 | 28.47 | 0.7985 | 0.2680 | 0.2126 | 30.04 | 0.8224 | 0.3381 | 0.2524 |

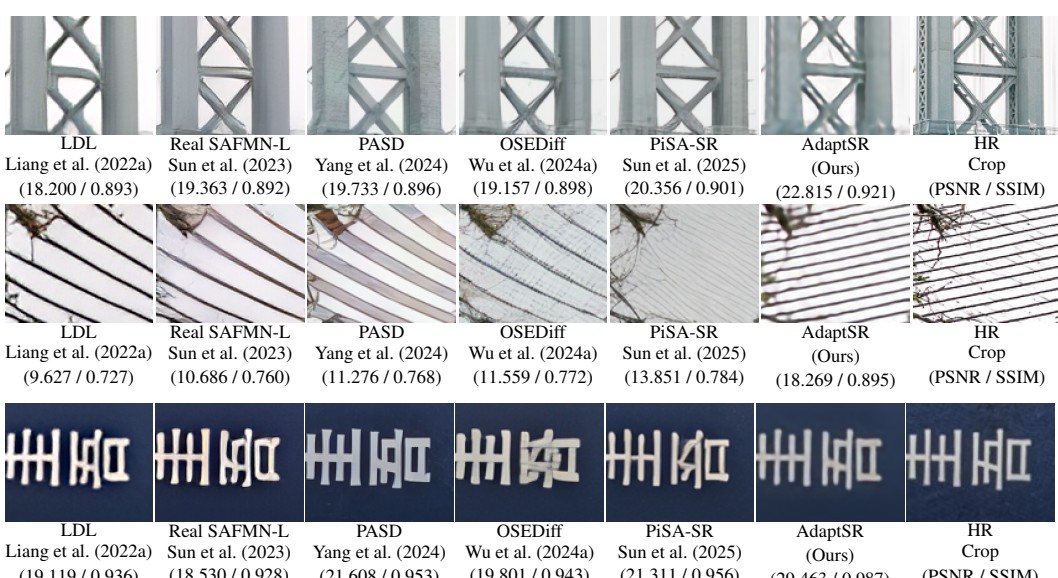

| LDL | Real SAFMN-L | PASD | OSEDiff | PiSA-SR | AdaptSR | HR |
|---|---|---|---|---|---|---|
| Liang et al. (2022a) | Sun et al. (2023) | Yang et al. (2024) | Wu et al. (2024a) | Sun et al. (2025) | (Ours) | Crop |
| (18.200 / 0.893) | (19.363 / 0.892) | (19.733 / 0.896) | (19.157 / 0.898) | (20.356 / 0.901) | (22.815 / 0.921) | (PSNR / SSIM) |

| LDL | Real SAFMN-L | PASD | OSEDiff | PiSA-SR | AdaptSR | HR |
|---|---|---|---|---|---|---|
| Liang et al. (2022a) | Sun et al. (2023) | Yang et al. (2024) | Wu et al. (2024a) | Sun et al. (2025) | (Ours) | Crop |
| (9.627 / 0.727) | (10.686 / 0.760) | (11.276 / 0.768) | (11.559 / 0.772) | (13.851 / 0.784) | (18.269 / 0.895) | (PSNR / SSIM) |

| LDL | Real SAFMN-L | PASD | OSEDiff | PiSA-SR | AdaptSR | HR |
|---|---|---|---|---|---|---|
| Liang et al. (2022a) | Sun et al. (2023) | Yang et al. (2024) | Wu et al. (2024a) | Sun et al. (2025) | (Ours) | Crop |
| (19.119 / 0.936) | (18.530 / 0.928) | (21.608 / 0.953) | (19.801 / 0.943) | (21.311 / 0.956) | (29.463 / 0.987) | (PSNR / SSIM) |

Figure 3: Visual comparison of the proposed AdaptSR with the state-of-the-art methods for ×4 real SR. GAN and diffusion models fail to capture the correct content of images, exhibiting excessive sharpness and color shifts. In contrast, our LoRA-based models reconstruct high-fidelity details with correct alignment, especially in complex areas with structured patterns. *Further visual comparisons are available in the Supplementary Materials.*

lucinate textures for perceptual realism, whereas AdaptSR prioritizes faithful reconstruction under real degradations. Consequently, while our perceptual scores are lower on synthetic DIV2K—where LPIPS/DISTS reward hallucinated detail—AdaptSR attains stronger or comparable perceptual quality on real datasets (RealSR, DRealSR), where such hallucination harms fidelity. To conclude, our lightweight LoRA-based method consistently outperforms both GAN and diffusion-based SoTA approaches across distortion metrics (PSNR, SSIM) while achieving competitive perceptual scores (LPIPS, DISTS) with minimal parameter overhead. This establishes AdaptSR as an efficient and scalable solution for real-world SR, bridging the gap between synthetic and real training without excessive computational costs.

**Qualitative Performance.** Figure 3 illustrates visual comparisons among ×4 real SR methods, emphasizing the strengths of our AdaptSR approach. The qualitative results align closely with quantitative outcomes, revealing that GAN-SR models like LDL Liang et al. (2022a) and Real SAFMN-Large Sun et al. (2023) show excessive sharpness with content inaccuracies such as misaligned

| LR | SwinIR | PASD | Ours | HR |
| bicubic | 11M | 625M | 886k | Params. |

Figure 4: AdaptSR reconstructs the correct digit ("6") while the SoTA PASD model hallucinates it as "K" on a DSLR Ignatov et al. (2017) iPhone image.

Table 2: Rank analysis on RealSR Cai et al. (2019) for 100k iterations with SwinIR. Uniform ranks are swept across values $r \in \{1, 4, 8, 16, 64\}$. Rank-aware allocation achieves the best trade-off under the same ~8% parameter budget.

| Model | PSNR | SSIM | LPIPS | DISTS | Updated Parameters |
|---|---|---|---|---|---|
| Baseline | 27.5537 | 0.7738 | 0.3965 | 0.2426 | 12M |
| + FineTuning | 28.2637 | 0.7922 | 0.3016 | 0.2274 | +12M |
| $r = 1$ | 28.3383 | 0.7957 | 0.2772 | 0.2129 | +152k (1.4%) |
| $r = 4$ | 28.2675 | 0.7945 | 0.2721 | **0.2125** | +466k (4.2%) |
| $r = 8$ | 28.3999 | 0.7976 | 0.2705 | 0.2128 | +886k (8.1%) |
| $r = 16$ | 28.3575 | 0.7959 | 0.2719 | 0.2140 | +1.7M (15.5%) |
| $r = 64$ | 28.3537 | 0.7959 | 0.2716 | 0.2130 | +6.8M (61.8%) |
| Rank-Aware | **28.4685** | **0.7985** | **0.2680** | 0.2126 | +886k (8.1%) |

striped patterns on roofs. Similarly, diffusion methods like PASD Yang et al. (2024) and OSEDiff Wu et al. (2024a) introduce artifacts with inaccurate orientations for the first image. In contrast, our AdaptSR method successfully reconstructs fine details, accurately aligning roof stripes. Furthermore, in the second image, LDL Liang et al. (2022a) and Real SAFM-L Sun et al. (2023) incorrectly render structure and colors. Likewise, diffusion methods PASD Yang et al. (2024) and OSEDiff Wu et al. (2024a) produce inaccurate semantic generation. On the contrary, AdaptSR preserves the correct structure and colors in the second case. Overall, our approach effectively controls artifacts while maintaining image fidelity, producing photorealistic, high-quality results in domain adaptation from bicubic to real SR.

### 4.3 GENERALIZATION: RAPID ADAPTATION ON *iPhone* IMAGES

To assess AdaptSR's real-world applicability, we fine-tuned the model on *iPhone* images from the DSLR Ignatov et al. (2017) dataset, which presents authentic, device-specific degradations. Notably, AdaptSR required less than 1 hour of training on a single GPU to effectively adapt to the characteristics of this new domain. Despite this minimal update, AdaptSR successfully captured the degradation patterns of the *iPhone* camera pipeline, yielding perceptually accurate reconstructions. Figure 4 showcases a notable example where AdaptSR accurately reconstructs fine details—such as the number "6" on a tool dial—that were misrepresented as a "K" by PASD Yang et al. (2024), a state-of-the-art model with 625M parameters. These findings highlight AdaptSR's practical value for scalable deployment, especially in scenarios requiring quick specialization without sacrificing reconstruction fidelity for real-world SR applications across diverse mobile imaging devices.

### 4.4 UNIFORM VS. RANK-AWARE RANK ANALYSIS

To complement our adaptive strategy, we first analyze the effect of fixed uniform ranks by sweeping $r \in \{1, 2, 4, 8, 16, 64\}$. Table 2 reports PSNR, SSIM, LPIPS, and DISTS on RealSR Cai et al. (2019) for ×4 images using SwinIR Liang et al. (2021). Even at $r = 1$ (only 1.4% of trainable parameters), performance closely matches full fine-tuning with 12M parameters, confirming that low-rank updates are effective. Optimal fidelity and perceptual quality are achieved around $r = 8$, with higher ranks offering no significant gains despite a steep increase in parameter count. These results support prior findings Zeng & Lee (2023) that pretrained models can be adapted effectively

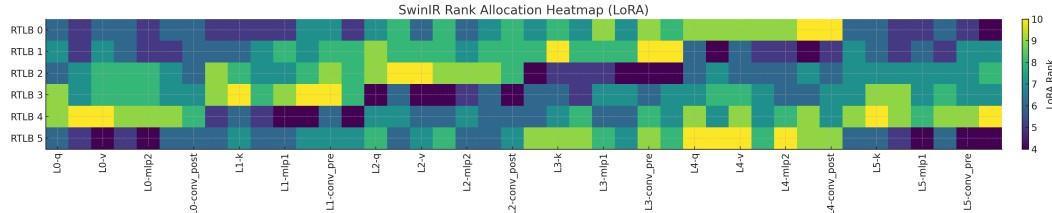

Figure 5: The heatmap visualizes LoRA ranks across residual transformer blocks (rows) and per-layer modules (columns). Rank-aware allocation concentrates capacity on later blocks and attention/MLP projections, while leaving early or less critical modules at minimal ranks. This focused distribution highlights the structural priorities of SR models and explains why rank-aware allocation outperforms uniform assignment under the same budget.

Table 3: PEFT methods vs. AdaptSR comparison on RealSR Cai et al. (2019). The best and the second-best results are highlighted in red and blue, respectively.

| Model | PSNR | SSIM | LPIPS↓ | Updated / Inference Params. |
|---|---|---|---|---|
| Baseline Liang et al. (2021) | 27.55 | 0.7738 | 0.3965 | - / 12M |
| Fine-Tuning | 28.26 | 0.7922 | 0.3016 | +12M / +12M |
| ARC (svd50) Dong et al. (2024) | 28.08 | 0.7894 | 0.2995 | +125k / +125k |
| ARC (svd100) | 28.08 | 0.7933 | 0.2883 | +236k / +236k |
| ARC (svd350) | 28.22 | 0.7954 | 0.2735 | +794k / +794k |
| ARC (svd375) | 28.37 | **0.7993** | 0.2714 | +850k / +850k |
| ARC (svd400) | 28.09 | 0.7924 | 0.2760 | +906k / +906k |
| ARC (svd500) | 28.16 | 0.7949 | 0.2711 | +1.13M / +1.13M |
| DiffFit Xie et al. (2023a) | 28.27 | 0.7949 | 0.2949 | +13k / +13k |
| RaSA He et al. (2025) | 28.18 | 0.7942 | 0.2790 | +7100k / +7100k |
| SHiRA Bhardwaj et al. (2024) | 28.31 | 0.7961 | 0.2842 | +930k / +930k |
| AdaptSR (Ours) | **28.47** | 0.7985 | **0.2680** | +886k / - |

with low-rank updates. We then compare uniform rank assignment to our proposed rank-aware allocation, where ranks are distributed based on gradient importance under the same parameter budget (∼8%). As shown in Table 2, rank-aware allocation achieves superior PSNR/SSIM and lower LPIPS/DISTS compared to uniform $r = 8$. This validates that concentrating capacity on later transformer blocks and key attention/MLP projections leads to better utilization of the parameter budget. Together with the heatmap visualization in Figure 5, these results demonstrate that rank-aware allocation captures meaningful structural priorities in SR architectures and yields stronger adaptation than uniform strategies. A five-seed evaluation further confirms that this improvement is statistically significant: rank-aware allocation exceeds uniform $r=8$ by $0.07$ dB on average with a standard deviation of $0.015$ dB ($p < 0.01$) under identical parameter budgets, please refer to Table 12.

## 4.5 COMPARISON WITH OTHER PEFT METHODS

Table 3 compares AdaptSR (LoRA-based) with other PEFT methods on RealSR Cai et al. (2019). AdaptSR achieves a PSNR of 28.40 dB, surpassing most ARC Dong et al. (2024) variants and DiffFit Xie et al. (2023a) while using fewer trainable parameters than ARC (svd500) and standard fine-tuning. Unlike ARC, which introduces additional inference overhead, AdaptSR seamlessly integrates into the base model post-training, preserving the original inference efficiency. Furthermore, AdaptSR consistently delivers strong perceptual quality, achieving the lowest LPIPS score, making it a more efficient and effective alternative for realSR adaptation.

## 4.6 DISCUSSION AND ABLATION STUDIES

**AdaptSR vs. Fine-Tuning.** Another key advantage of AdaptSR is its rapid convergence. As shown in Figure 6, AdaptSR reaches peak performance within 100k iterations—approximately 4 hours for SwinIR and 1 hour for SAFMN on an NVIDIA RTX 4090—compared to 23 and 9 hours, respectively, for fine-tuning. Importantly, since LoRA layers merge into the base model post-training, the inference speed remains unchanged, making AdaptSR a practical and highly efficient alternative to FT for domain adaptation.

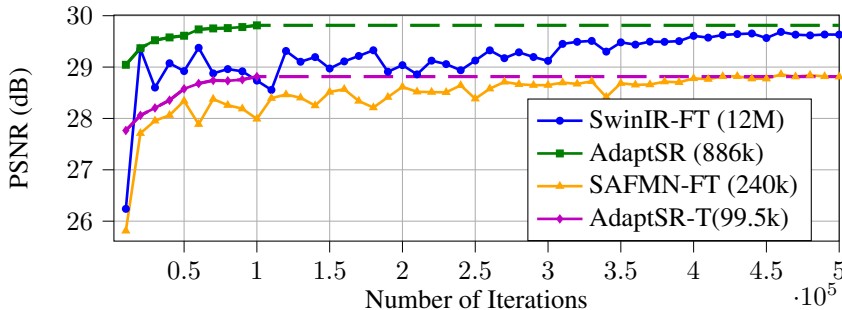

Figure 6: Number of iterations vs. performance comparison for full model finetuning and our AdaptSR approach for SwinIR Liang et al. (2021) and SAFMN Sun et al. (2023) on DIV2K Unknown Validation Set Agustsson & Timofte (2017). LoRA reaches peak PSNR in just 100k iterations, completing in 4 hours for SwinIR and 1 hours for SAFMN on an NVIDIA RTX 4090, whereas fine-tuning takes significantly longer—23 hours for SwinIR and 9 hours for SAFMN—to achieve similar performance. The dashed lines of LoRA methods indicate their final PSNR performance at their latest iteration 100k.

Table 4: *Comparison of AdaptSR's adaptive merging and full fine-tuning across modules for domain adaptation on DRealSR Wei et al. (2020).*

|  | Convs | | MLPs | | MSAs | |
|---|---|---|---|---|---|---|
|  | FT | LoRA | FT | LoRA | FT | LoRA |
| Params. | 2.2M | 868k | 4.7M | 1.2M | 4.7M | 1.2M |
| PSNR | 29.29 | **29.77** | 29.82 | **29.92** | **29.83** | 29.77 |
| LPIPS | 0.3955 | **0.3924** | 0.3457 | **0.3399** | **0.3465** | 0.4023 |

**Module-wise Adaptive Merging Scheme vs. FT.** Table 4 compares our adaptive merging strategy in AdaptSR with full fine-tuning (FT) on DRealSR Wei et al. (2020), across three core modules: convolutional layers (Convs), MLPs, and multi-head self-attention (MSA). Results show that AdaptSR achieves comparable or better performance with significantly fewer updated parameters. In Convs, AdaptSR updates only 868k parameters (vs. 2.2M in FT) and achieves higher PSNR, effectively capturing low-level textures. For MLPs, it improves PSNR with just 1.2M parameters (vs. 4.7M in FT), with a slight drop in perceptual quality. In MSAs, it maintains similar performance with notable parameter savings. These results highlight that AdaptSR's efficiency gains are most pronounced in Convs and MLPs. Overall, the adaptive merging strategy offers a compact, efficient alternative to FT and serves as a practical guide for adapting transformer-based SR models under resource constraints.

**Sustainability and Efficiency.** By combining selective LoRA integration with rank-aware allocation, AdaptSR trains only $8\%$ of parameters on average, converges $5$–$6\times$ faster than full fine-tuning, and adds *zero inference overhead*. This makes AdaptSR a practical solution for sustainable AI: lowering GPU hours, reducing energy usage, and enabling affordable domain adaptation even on constrained hardware.

## 5 CONCLUSION

We presented AdaptSR, a rank-aware low-rank adaptation framework that efficiently repurposes bicubic-trained SR models for real-world degradations. Unlike prior LoRA applications, AdaptSR integrates architecture-specific design with a gradient-based rank allocation strategy, ensuring that limited adapter capacity is concentrated on the most impactful layers. This enables up to 92% parameter reduction and days-to-hours speedups, while guaranteeing identical inference cost after merging. Extensive experiments across CNN and Transformer backbones confirm that AdaptSR consistently matches or surpasses full fine-tuning and outperforms state-of-the-art GAN and diffusion methods in distortion metrics, while offering competitive perceptual quality. By unifying efficiency, sustainability, and scalability, AdaptSR provides a practical and budget-friendly path for adapting

SR models to diverse real-world degradations, paving the way for deployment on lightweight and resource-constrained devices.

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
