## A    APPENDIX - ADAPTSR: RANK-AWARE LOW-RANK ADAPTATION FOR REAL-WORLD SUPER-RESOLUTION

This document provides additional comparisons of fine-tuning and AdaptSR in terms of training efficiency, memory consumption and inference time in Section B. Furthermore, we compare domain adaptation between baseline models and our LoRA-enhanced AdaptSR versions in Section D. Section E highlights the module-wise significance of low-rank adaptation for AdaptSR-C, a CNN-based architecture. Lastly, Section H offers visual comparisons between GAN approach and our AdaptSR-GAN for real SR, while Section I presents additional visual comparisons between our methods and the state-of-the-art GAN and diffusion models.

## B    ADAPTSR VS. FINE-TUNING COMPARISON

To evaluate the generality of our LoRA-based adaptation, we apply it across diverse SR architectures: SwinIR Liang et al. (2021) (AdaptSR), SAFMN Sun et al. (2023) (AdaptSR-T), EDSR Lim et al. (2017) (AdaptSR-C), and DRCT Hsu et al. (2024) (AdaptSR-L). Table **??** compares bicubic-trained models, full fine-tuning (FT), and our LoRA-based variants on RealSR Cai et al. (2019). AdaptSR consistently matches or surpasses FT while drastically reducing trainable parameters, memory, and training time. For example, on SwinIR, AdaptSR outperforms FT (+0.42 dB PSNR, +2% SSIM, –3.2% LPIPS) using only 886k parameters and 4 hours of training, compared to 12M and 23 hours. On SAFMN, AdaptSR-T matches FT's 0.78 dB gain with half the parameter cost. Even in large models like DRCT, AdaptSR-L achieves comparable performance with 6× fewer parameters and 6× faster training. Perceptual gains are confirmed by LPIPS and DISTS, highlighting AdaptSR's effectiveness across model scales.

Figure 7 compares AdaptSR to full fine-tuning for bicubic-trained SR models. Fine-tuning updates all parameters, increasing memory usage and storage demands, while AdaptSR selectively updates lightweight LoRA layers for efficient adaptation. Full fine-tuning requires backpropagation through the entire network, leading to high GPU memory usage, especially for large transformer-based SR models. In contrast, AdaptSR reduces memory consumption by over 92% by freezing the pretrained backbone. Additionally, AdaptSR merges LoRA updates post-training, introducing no inference cost. Storage-wise, full fine-tuning doubles model size, whereas AdaptSR only saves LoRA layers (e.g., 886k vs. 12M parameters), making it ideal for deployment on storage-constrained devices.

## C    ADAPTIVE MERGING FOR EFFICIENT ADAPTATION

To determine the most critical modules for adapting Transformer architecture (i.e. SwinIR Liang et al. (2021)) from bicubic to real SR, we experimented with merging LoRA in four configurations: (1) all layers (convolutional, linear, and attention), (2) only convolutional layers, (3) only MSA layers, and (4) only MLP layers. As shown in Figure 8, applying LoRA to all layers achieved full fine-tuning performance in just 50 minutes (compared to 23 hours) while using only 8% of the parameters. Notably, LoRA applied exclusively to convolutional layers achieved nearly identical performance within 30 minutes, requiring just 2% of the parameters. Regardless of configuration, all methods started from 22.60 dB PSNR, matching the bicubic baseline.

To further analyze the contributions of different modules, we employed Local Attribution Maps (LAM) Gu & Dong (2021), which highlight feature interactions and contextual dependencies. As visualized in Figure 9, applying LoRA across all layers enhances feature integration beyond individual module-based adaptations or full fine-tuning. Additionally, Diffusion Index (DI) Gu & Dong (2021) values confirm that LoRA applied to all layers achieves the broadest attention range and strongest reconstruction quality, followed closely by the convolution-only configuration. Interestingly, even MLP-only and attention-only LoRA surpass standard fine-tuning in feature capture ability. This analysis demonstrates that LoRA applied to all layers is the optimal strategy for real SR adaptation, offering a balance between efficiency and reconstruction quality. However, for highly constrained settings, focusing solely on convolutional layers (2% parameters) provides an efficient yet competitive alternative.

## Low-Rank Adaptation (LoRA) vs. Full Model Fine-Tuning

**During Training**

❄️Pretrained Weights (Bicubic SR) 12M

🔥LoRA Weights

886k

🔥Pretrained Weights (Bicubic SR) 12M

**Inference Time**

Merged Weights (Real SR) 12M

Superior Performance

Fine-Tuned Weights (Real SR) 12M

**Storage for Generalization**

Bicubic SR Weights (12M)

Real SR Weights (886k)

Bicubic SR Weights (12M)

Real SR Weights (12M)

Figure 7: **LoRA vs. FT:** Our efficient LoRA approach adapts SR models from bicubic to real-world degradations by updating only 8% of parameters, avoiding the doubled storage requirements of full fine-tuning with no extra inference cost, while achieving comparable or superior performance on RealSR Cai et al. (2019).

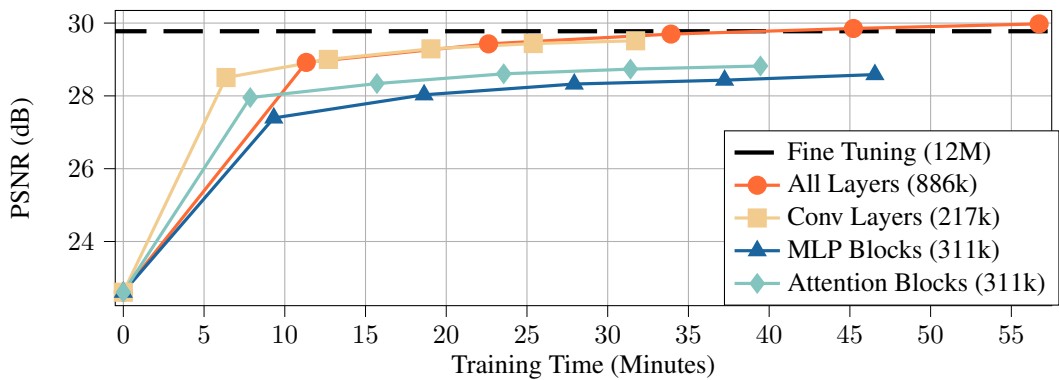

Figure 8: Training time vs. performance comparison for different adaptation modules with 100k iterations on DIV2K unknown Agustsson & Timofte (2017). The dashed line indicates the best PSNR achieved by standard fine-tuning, which requires 4 days of training. Notably, LoRA applied to all layers reaches this PSNR level in just 30 minutes.

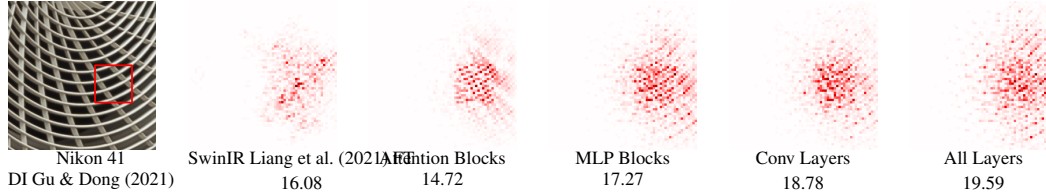

| Nikon 41 | SwinIR Liang et al. (2021) | Attention Blocks | MLP Blocks | Conv Layers | All Layers |
| DI Gu & Dong (2021) | 16.08 | 14.72 | 17.27 | 18.78 | 19.59 |

Figure 9: Comparison of local attribution maps (LAMs) Gu & Dong (2021) between different adaptation modules and the standard FT model on a RealSR-Nikon41 Cai et al. (2019) image. LAMs indicate the importance of each pixel in reconstructing the patch marked by the red box. Higher diffusion index (DI) Gu & Dong (2021) values suggest broader contextual usage and potentially better reconstruction quality.

Table 5: *Effect of CNN-LoRA layers for domain adaptation* from bicubic to real SR on DRealSR Wei et al. (2020) dataset for rank 32.

| LoRA Layer | PSNR | SSIM | LPIPS | Trainable Parameters | Training Time (Mins) |
|---|---|---|---|---|---|
| MSAs | 30.771 | 0.8402 | 0.4023 | 1.2M | 77 |
| MLPs | **31.024** | 0.8433 | 0.3999 | 1.2M | 57.71 |
| All Convs | 30.974 | **0.8425** | **0.3994** | 868k | 31.45 |
| First Conv | 30.078 | 0.8226 | 0.4327 | 53k | 24.95 |
| RSTLB Convs | 30.968 | 0.8422 | 0.4003 | 622k | 41.09 |
| DFE Convs | 29.530 | 0.8088 | 0.4470 | 104k | 19.22 |
| BU Conv | 29.583 | 0.8117 | 0.4379 | 70k | 19.02 |
| AU Conv | 30.479 | 0.8305 | 0.4201 | 19k | 18.42 |

**Effect of CNN-LoRA Layers.** Table 5 shows the impact of layer-specific updates for adapting SwinIR Liang et al. (2021) to real SR. Partitioning and evaluating convolutional layers reveals that certain layers, especially those in residual blocks (RSTLB Conv) and the final convolution layer after upsampling (AU Conv), achieve higher fidelity and perceptual scores with fewer parameter updates than MSA and MLP layers on DRealSR Wei et al. (2020) at rank 32. Notably, the AU Conv layer delivers the highest performance gains, outperforming both the initial convolution layer (First Conv), the ones in the deep feature extraction module (DFE Convs) and pre-upsampling layer (BU Conv). Overall, adapting MLP linear layers or AU Conv optimally supports domain adaptation.

## C.1 LAYER SORTING IN RANK-AWARE ALLOCATION

To determine which layers are most critical for adaptation, we compute the $\ell\_2$-norm of the gradient of the loss with respect to each trainable layer:

$$s_\ell = \|\nabla_{W_\ell}\mathcal{L}(\mathcal{B})\|_2$$

Here, $\mathcal{B}$ is a mini-batch of training data, $\mathcal{L}$ is the loss function (e.g., $\ell\_1$ or perceptual), and $W\_\ell$ is the weight matrix of layer $\ell$. This score reflects the sensitivity of the loss to changes in each layer. After computing $s_\ell$ for all LoRA-eligible layers, we sort the layers in descending order of importance:

$$\text{Sort } \{W_\ell\} \text{ such that } s_{\ell_1} \geq s_{\ell_2} \geq \cdots \geq s_{\ell_L}$$

This sorted order guides rank assignment: more important layers receive higher ranks, while less important ones may receive a smaller rank or be skipped entirely depending on the parameter budget.

We also evaluated whether layer importance shifts during training by recomputing gradient-based scores at later checkpoints (3k and 6k iterations) and presented results in Table 6. The resulting importance rankings remained highly stable (Spearman $\rho > 0.88$), indicating that early gradients capture the dominant adaptation structure. Reallocating ranks based on these intermediate scores changed PSNR by less than 0.02 dB, suggesting minimal benefit from dynamic reallocation. This stability is consistent with prior observations in pruning and sensitivity-analysis literature (e.g., SNIP Lee et al.

| Method | PSNR |
|---|---|
| dynamic@$3k$ | 28.46 |
| dynamic@$6k$ | 28.45 |
| static one-shot | 28.47 |

Table 6: Dynamic reallocation by recomputing importance at 3k and 6k iterations vs. static reallocation AdaptSR.

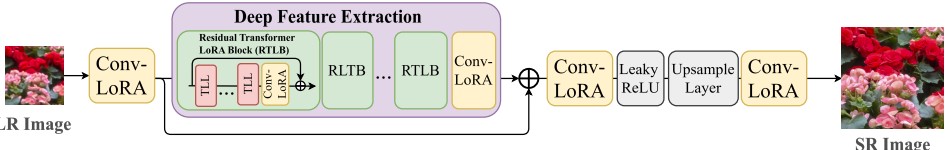

Figure 10: Overview of AdaptSR. We inject rank-8 LoRA adapters (orange) into both convolution (Conv-LoRA) and transformer (MSA/MLP-LoRA) layers of a bicubic-trained SR backbone (SwinIR shown as an example). Only the low-rank matrices A and B are updated; after training they are merged back, so inference cost is unchanged. The same plug-and-play recipe applies unchanged to any CNN- or Transformer-based SR network.

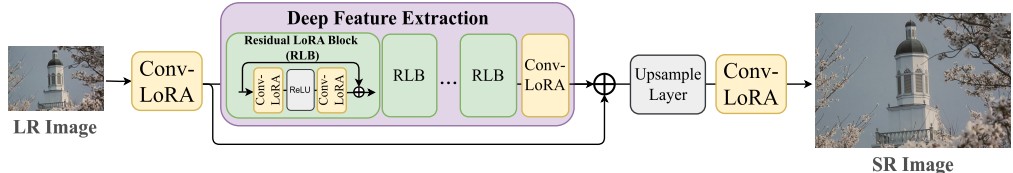

Figure 11: Overview of AdaptSR-C, CNN based on EDSR Lim et al. (2017) to adapt bicubic-trained SR models to real-world degradations. The deep feature extraction block includes 16 Residual LoRA Blocks (RLB). LoRA-modified convolutional reduce parameters and computational load, enabling efficient, high-resolution outputs.

(2019), GraSP Wang et al. (2020)), where initial gradients provide the most informative saliency cues.

## D    MORE COMPARISON FOR DOMAIN ADAPTATION

Table 7 shows additional comparisons between baseline models and our AdaptSR versions for domain adaptation from bicubic to real-world SR on DIV2K Agustsson & Timofte (2017) and DRealSR Wei et al. (2020). Pretrained bicubic SR models exhibit notable performance drops when evaluated on unknown real-world images with complex degradations, representing their limited adaptability. Our LoRA-based method effectively addresses this gap, significantly enhancing performance on both datasets. Specifically, on the DIV2K validation set, our LoRA-enhanced models achieve up to 1 dB improvement in PSNR and a 25% reduction in LPIPS, reflecting both higher fidelity and perceptual quality. Similarly, on the challenging DRealSR dataset, LoRA adaptation boosts PSNR by 0.5 dB and reduces LPIPS by 20%, despite not utilizing the DRealSR training set. This underscores the strong generalization capability of our approach, enabling it to adapt pretrained bicubic models to diverse real-world degradations effectively and efficiently.

## E    MODULE-WISE IMPORTANCE FOR ADAPTSR-CNN

We provide module-wise importance for AdaptSR-Transformer and AdaptSR-CNN architectures presented in Figure 10 and 11, respectively for domain adaptation from bicubic to real SR, we tested LoRA on four configurations: (1) first convolution layer (2) convolutional layers in Residual Blocks, (3) convolution layer before upsampling, and (4) convolution layer after upsampling. As presented in Table 8, applying LoRA to all convolutional layers updating only 314k parameters of

Table 7: Performance comparison for domain adaptation between bicubic-trained baseline models and LoRA-enhanced AdaptSR versions on realSR benchmarks Agustsson & Timofte (2017); Cai et al. (2019); Wei et al. (2020). LoRA substantially enhances validation results, achieving effective domain adaptation from bicubic to real SR with significantly fewer trainable parameters and reduced training time, while preserving the inference speed of the baseline models.

| | Metric | EDSR | AdaptSR-C | SAFMN | AdaptSR-T | SwinIR | AdaptSR | DRCT | AdaptSR-L |
|---|---|---|---|---|---|---|---|---|---|
| | PSNR ↑ | 24.9164 | 25.5393 | 24.9298 | 25.4742 | 24.8738 | **25.8058** | 24.8938 | 25.7739 |
| DIV2K | SSIM ↑ | 0.6261 | 0.6553 | 0.6264 | 0.6547 | 0.6237 | 0.6681 | 0.6249 | **0.6693** |
| | LPIPS ↓ | 0.6740 | 0.5376 | 0.6758 | 0.5461 | 0.6731 | 0.5047 | 0.6744 | **0.5037** |
| | DISTS ↓ | 0.3304 | 0.3324 | 0.3308 | 0.3271 | 0.3305 | 0.3194 | 0.3301 | **0.3167** |
| | PSNR ↑ | 27.5422 | 27.9322 | 27.5383 | 28.3216 | 27.5537 | 28.7003 | 27.5502 | **28.7245** |
| RealSR | SSIM ↑ | 0.7732 | 0.7878 | 0.7740 | 0.7935 | 0.7738 | **0.8079** | 0.7737 | 0.8061 |
| | LPIPS ↓ | 0.3937 | 0.2835 | 0.3927 | 0.2935 | 0.3965 | 0.2591 | 0.3966 | **0.2553** |
| | DISTS ↓ | 0.2303 | 0.2190 | 0.2472 | 0.2291 | 0.2304 | 0.2109 | 0.2584 | **0.2094** |
| | PSNR ↑ | 30.6469 | 30.6370 | 30.6221 | **30.9162** | 30.6209 | 30.7392 | 30.6066 | 30.8109 |
| DRealSR | SSIM ↑ | 0.8328 | 0.8368 | 0.8327 | 0.8403 | 0.8327 | **0.8422** | 0.8325 | 0.8419 |
| | LPIPS ↓ | 0.4364 | 0.3523 | 0.4356 | 0.3529 | 0.4391 | **0.3381** | 0.4394 | 0.3414 |
| | DISTS ↓ | 0.2695 | 0.2548 | 0.2689 | 0.2545 | 0.2708 | **0.2524** | 0.2703 | 0.2544 |

Table 8: *Module-wise importance for LoRA domain adaptation* for AdaptSR-C architecture from bicubic to real SR on RealSR Cai et al. (2019) dataset for rank 8.

| LoRA Layer | PSNR | SSIM | LPIPS | DISTS | Trainable Parameters |
|---|---|---|---|---|---|
| All Convs | **27.93** | **0.7878** | **0.2835** | **0.2190** | 314k (21%) |
| First Conv | 26.55 | 0.7473 | 0.3773 | 0.2552 | 4.8k (0.3%) |
| RLB Convs | 27.54 | 0.7741 | 0.3935 | 0.2438 | 295k (20%) |
| BU Conv | 27.54 | 0.7740 | 0.3941 | 0.2437 | 9.2k (0.6%) |
| AU Conv | 27.54 | 0.7740 | 0.3937 | 0.2436 | 4.8k (0.6%) |

baseline model (1.5M) yield better fidelity and perceptual scores than individual convolutional layers. However, applying low-rank adaptation to convolutional layers either before upsampling (BU Conv) or after upsampling (AU Conv) yields better performance than adapting the initial convolutional layer (First Conv). Notably, these configurations achieve performance comparable to adapting convolutional layers within residual LoRA blocks (RLB Convs) while requiring significantly fewer parameter updates. Among these, the AU Conv layer stands out, providing optimal support for domain adaptation with an update of only 0.6k parameters.

**Rationale for the Conv-LoRA decomposition.** We adopt a $1\times1 \to k\times k$ (k=3) factorization for the convolutional LoRA update in Eq. (2) to efficiently capture both channel mixing and spatial structure. The $1\times1$ branch acts as a low-cost channel projection, while the subsequent $k\times k$ kernel injects spatial context. This design mirrors depthwise–separable decompositions used in efficient CNNs and aligns with observations that SR convolution kernels exhibit low intrinsic rank. To verify this choice, we evaluated alternative factorizations—including $k\times k \to 1\times1$, single-kernel LoRA (only $1\times1$ or only $k\times k$), and symmetric two-layer forms—under equal parameter budgets. As reported in Table 9, the proposed $1\times1 \to k\times k$ decomposition achieves the best accuracy–parameter trade-off, confirming that it provides the most effective low-rank basis for adapting spatial filters in SR models.

Table 9: Ablation of Conv-LoRA factorizations on RealSR using SwinIR for 100k iterations under an equal ∼8% parameter budget. The proposed $1\times1 \to k\times k$ design provides the best accuracy–parameter balance.

| Factorization | PSNR | SSIM | LPIPS↓ | Params |
|---|---|---|---|---|
| $1\times1$ only | 28.325 | 0.7961 | 0.2769 | 886k |
| $k\times k$ only | 28.352 | 0.7964 | 0.2745 | 886k |
| $k\times k \to 1\times1$ | 28.412 | 0.7970 | 0.2731 | 886k |
| $1\times1 \to k\times k$ (ours) | **28.468** | **0.7985** | **0.2680** | 886k |

Table 10: Ablation on scaling factor on RealSR Cai et al. (2019) for AdaptSR.

| Scaling Factor | PSNR | SSIM | LPIPS | DISTS |
|---|---|---|---|---|
| 1 | 27.6252 | 0.7767 | 0.3917 | 0.2430 |
| 0.5 | 27.6684 | 0.7780 | 0.3825 | 0.2414 |
| 0.25 | 27.7357 | 0.7801 | 0.3688 | 0.2395 |
| 0.125 | **28.7023** | **0.8079** | **0.2591** | **0.2109** |
| 0.0625 | 22.6364 | 0.6357 | 0.4246 | 0.3066 |

## F  RANK ANALYSIS FOR OTHER BASELINES

**Scaling Factor.** The scaling factor in AdaptSR plays a crucial role in balancing adaptation strength and stability during domain adaptation. As shown in Table 10, reducing the scaling factor from 1 to 0.125 consistently improves PSNR, SSIM, and perceptual metrics, indicating more effective adaptation to real-world degradations. Notably, a scaling factor of 0.125 achieves the best overall performance, suggesting that smaller updates help fine-tune the model without overfitting to specific degradations. However, an excessively low scaling factor (0.0625) leads to a sharp drop in PSNR and perceptual quality, likely due to insufficient adaptation capacity. These results highlight the importance of carefully selecting the scaling factor to optimize performance while maintaining stability in real SR adaptation.

## G  COMPARISON WITH ALTERNATIVE SCORING METRICS

We evaluate several alternatives under the same parameter budget: diagonal Fisher information, activation norms, and random allocation. The gradient-based criterion consistently achieves the highest PSNR and lowest LPIPS across RealSR with SwinIR backbone in Table 11. These results confirm both the theoretical suitability and empirical robustness of the gradient-norm criterion for rank allocation.

Table 11: Comparison of different layer-importance metrics for allocating LoRA ranks under an equal ∼8% parameter budget (886k). Gradient-based scoring yields the best fidelity (PSNR/SSIM) and perceptual (LPIPS) results. Evaluated on RealSR Cai et al. (2019) using SwinIR for 100k iterations.

| Importance Metric | PSNR | SSIM | LPIPS↓ | DISTS↓ |
|---|---|---|---|---|
| Fisher diagonal | 28.412 | 0.7971 | 0.2742 | 0.2134 |
| Activation norm | 28.391 | 0.7967 | 0.2761 | 0.2140 |
| Random allocation | 28.324 | 0.7949 | 0.2853 | 0.2162 |
| Gradient norm (ours) | **28.4685** | **0.7985** | **0.2680** | **0.2126** |

Table 12: Five-seed comparison between uniform $r=8$ and rank-aware allocation under identical ∼8% parameter budgets on RealSR. Rank-aware allocation yields a statistically significant improvement ($p < 0.01$).

| Method | PSNR (mean ± std) | SSIM (mean ± std) | LPIPS↓ (mean ± std) |
|---|---|---|---|
| Uniform $r=8$ | 28.3999 ± 0.017 | 0.7976 ± 0.0009 | 0.2705 ± 0.0021 |
| Rank-Aware (ours) | **28.4685 ± 0.015** | **0.7985 ± 0.0008** | **0.2680 ± 0.0017** |

**Extention to GAN-Based Real SR.** We further extend LoRA-based domain adaptation to GAN-based SR with AdaptSR-GAN, demonstrating its flexibility and efficiency. Using SwinIR Liang et al. (2021) with LoRA layers and a U-Net discriminator with spectral normalization Wang et al. (2021), we employ adversarial training with a weighted combination of L1, perceptual Johnson et al. (2016), and GAN losses (1:1:0.1) as in RealESRGAN. As shown in Table 13, AdaptSR-GAN outperforms RealESRGAN Wang et al. (2021) and LDL Liang et al. (2022a) in both fidelity and perceptual quality while requiring only 886k parameters—far fewer than the 12M in LDL and 17M in RealESRGAN. These results establish AdaptSR-GAN as an efficient alternative for GAN-based SR, effectively handling real-world degradations.

Table 13: Comparison of GAN-based and fidelity-oriented RealSR methods and our adversarially trained AdaptSR-GAN (with SwinIR Liang et al. (2021) baseline) on DIV2K Agustsson & Timofte (2017) validation patches. Our method achieves superior fidelity and perceptual quality with significantly fewer parameters.

| Model | Params. | PSNR ↑ | SSIM ↑ | LPIPS ↓ | DISTS ↓ |
|---|---|---|---|---|---|
| EDSR-GAN | 1.5M | 21.62 | 0.5521 | 0.3984 | 0.2827 |
| EDSR + AdaptSR-GAN | +314k | **22.18** | **0.5715** | **0.3612** | **0.2649** |
| RealESRGAN | 17M | 21.94 | 0.5736 | 0.3868 | 0.2601 |
| ESRGAN + AdaptSR-GAN | 577k | 22.88 | 0.6202 | 0.3453 | 0.2588 |
| SwinIR-GAN | 12M | 22.94 | 0.6142 | 0.3457 | 0.2413 |
| LDL (RealSwinIR) | 12M | 23.76 | 0.6403 | 0.3091 | 0.2189 |
| SwinIR + AdaptSR-GAN (Ours) | 886k | **24.11** | **0.6598** | **0.2914** | **0.2185** |

Table 14: Performance comparison under two hybrid degradations—(i) ×4 downsampling with Gaussian noise ($\sigma = 30$) and (ii) ×4 downsampling with JPEG compression (quality factor = 30)—across five benchmarks. Most PETL methods exhibit notable degradation under these heterogeneous conditions, whereas AdaptSR maintains strong generalization and restoration quality.

| Hybrid Degradation | Model | Trainable Params. | Set5 PSNR | Set5 SSIM | Set14 PSNR | Set14 SSIM | BSD100 PSNR | BSD100 SSIM | Urban100 PSNR | Urban100 SSIM | Manga109 PSNR | Manga109 SSIM |
|---|---|---|---|---|---|---|---|---|---|---|---|---|
| LR4 & Noise30 | Baseline Liang et al. (2021) | 12M | 19.52 | 0.3421 | 19.02 | 0.2981 | 18.85 | 0.2654 | 18.27 | 0.3113 | 19.44 | 0.3680 |
| | VPT Jia et al. (2022) | 884K | 23.82 | 0.5420 | 22.71 | 0.4601 | 22.64 | 0.4210 | 21.01 | 0.4395 | 22.34 | 0.5425 |
| | Adapter Houlsby et al. (2019) | 691K | 25.32 | 0.6701 | 23.89 | 0.5720 | 23.91 | 0.5351 | 21.82 | 0.5498 | 23.29 | 0.6742 |
| | LoRA Hu et al. (2022) | 995K | 24.98 | 0.6204 | 23.61 | 0.5290 | 23.58 | 0.4922 | 21.59 | 0.5074 | 23.01 | 0.6241 |
| | AdaptFormer Chen et al. (2022a) | 677K | 25.84 | 0.7012 | 24.31 | 0.5951 | 24.18 | 0.5598 | 22.31 | 0.5821 | 24.10 | 0.7120 |
| | AdaptIR Guo et al. (2024) | 697K | 26.48 | 0.7441 | 24.88 | 0.6345 | 24.67 | 0.6279 | 22.88 | 0.5932 | 24.96 | 0.7625 |
| | AdaptSR | 886k | 26.62 | 0.7489 | 24.88 | 0.6330 | 24.58 | 0.5881 | 23.08 | 0.6444 | 25.55 | 0.7812 |
| LR4 & JPEG30 | Baseline Liang et al. (2021) | 12M | 24.82 | 0.6482 | 23.61 | 0.5681 | 21.28 | 0.5401 | 23.79 | 0.5431 | 22.31 | 0.6478 |
| | VPT Jia et al. (2022) | 884K | 26.14 | 0.7215 | 24.54 | 0.6159 | 22.18 | 0.5971 | 24.47 | 0.5741 | 23.71 | 0.7210 |
| | Adapter Houlsby et al. (2019) | 691K | 26.72 | 0.7540 | 25.07 | 0.6372 | 23.05 | 0.6428 | 24.74 | 0.5899 | 24.82 | 0.7701 |
| | LoRA Hu et al. (2022) | 995K | 26.75 | 0.7551 | 25.09 | 0.6384 | 23.03 | 0.6410 | 24.72 | 0.5884 | 24.78 | 0.7680 |
| | AdaptFormer Chen et al. (2022a) | 677K | 26.79 | 0.7562 | 25.12 | 0.6401 | 23.10 | 0.6433 | 24.73 | 0.5897 | 24.85 | 0.7715 |
| | AdaptIR Guo et al. (2024) | 697K | 27.13 | 0.7739 | 25.44 | 0.6545 | 23.41 | 0.6620 | 25.04 | 0.6057 | 25.29 | 0.7903 |
| | AdaptSR | 886k | 27.01 | 0.7703 | 25.55 | 0.6568 | 23.55 | 0.6668 | 25.02 | 0.6679 | 25.20 | 0.7884 |

## G.1 COMPARISON ON HYBRID DEGRADATION TASKS

To provide a more challenging and realistic evaluation, we additionally benchmark all methods under hybrid degradations that combine heterogeneous corruption sources. Using the SwinIR Liang et al. (2021) backbone, we compare AdaptSR against strong PETL baselines, including: (i) a frozen pretrained model, (ii) VPT Jia et al. (2022) (prompt-based adaptation), (iii) Adapter Houlsby et al. (2019) (bottleneck modules inserted after attention/MLP), (iv) LoRA Hu et al. (2022) (low-rank updates to attention projections), (v) AdaptFormer Chen et al. (2022a) (parallel MLP adapters), and (vi) AdaptIR Guo et al. (2024) (MoE-based spatial–channel adaptation). Table 14 reports results for two second-order hybrid settings: (i) LR4 & Noise30, defined as ×4 downsampling followed by Gaussian noise with $\sigma = 30$, and (ii) LR4 & JPEG30, defined as ×4 downsampling with JPEG compression at quality 30. These scenarios introduce mixed and frequency-disjoint distortions, making them a demanding test of adaptation robustness. As shown in Table 14, AdaptSR consistently achieves the best PSNR/SSIM across all five benchmarks. Under LR4 & Noise30, AdaptSR attains 23.08 dB on Urban100 and 25.55 dB on Manga109, outperforming AdaptIR by 0.20 dB and 0.59 dB, respectively. Under LR4 & JPEG30, AdaptSR again ranks first, exceeding AdaptIR by 0.11 dB on Set14 and 0.02 dB on Urban100, while updating fewer than 900k parameters. These results demonstrate that AdaptSR effectively models heterogeneous degradations and generalizes reliably under complex corruption mixtures.

## H VISUAL RESULTS FOR GAN-BASED LoRA

We present additional visual comparisons with state-of-the-art GAN method (LDL Liang et al. (2022a)) and our LoRA-based GAN method in Figure 13. It is worth mentioning that both architectures use the same generator architecture Liang et al. (2021) and in our adaptation approach we only injected LoRA layers to the same generator. Figure 13 shows that the proposed approach can further extend to adversarial training and provides clearer images with fine detailed structures than GAN method.

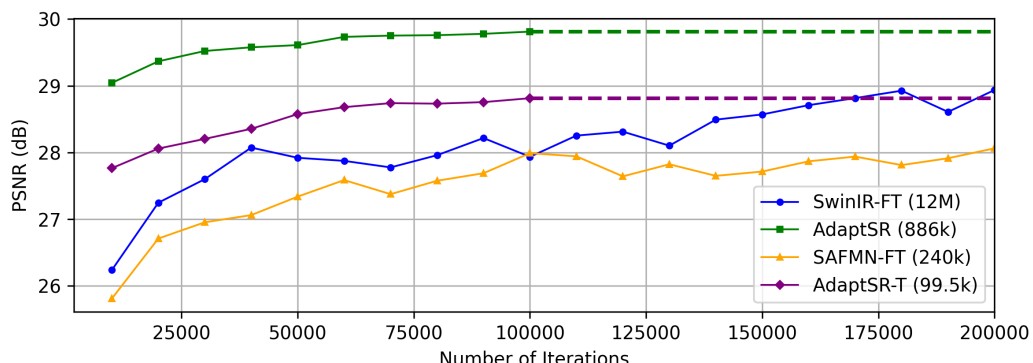

Figure 12: Comparison of iteration–performance trends for full-model finetuning (following the official RealSwinIR schedule Liang et al. (2021)) and our AdaptSR variants on SwinIR Liang et al. (2021) and SAFMN Sun et al. (2023). Finetuned models are trained for 200k iterations, while AdaptSR models follow our 100k training protocol (see Experimental Setup). Dashed lines for LoRA-based variants indicate their final PSNR reached at the last iteration (100k).

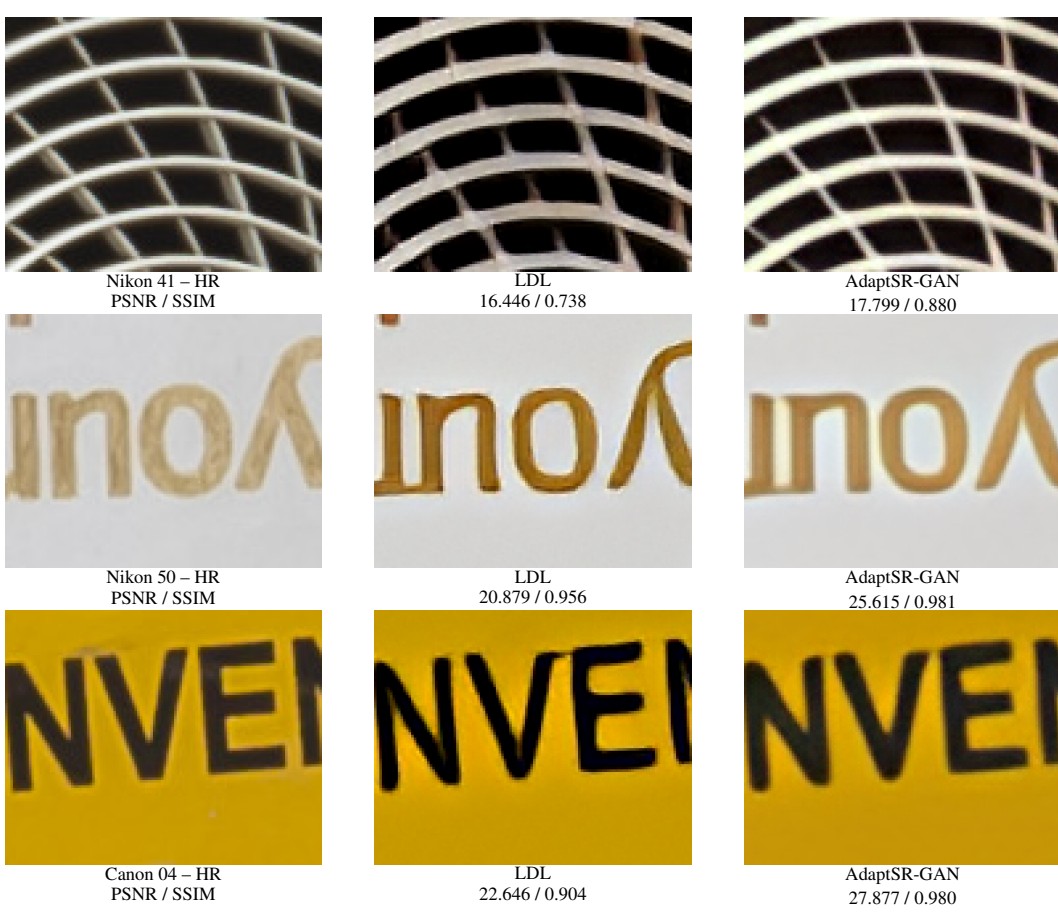

Figure 13: Visual results for GAN-based real SR methods on the RealSR Cai et al. (2019) dataset. The proposed model with adversarial training reconstructs sharper and more faithful textures.

# I  MORE VISUAL RESULTS

We provide further visual comparisons of ×4 Real SR results between our proposed AdaptSR method and the other state-of-the-art GAN and diffusion methods including LDL Liang et al. (2022a), Real-SAFMN-L Sun et al. (2023), PASD Yang et al. (2024) and OSEDiff Wu et al. (2024a) in Figure 14 to Figure 15. From these visual comparisons, one can draw consistent observations in line with the results in the paper. GAN models, LDL Liang et al. (2022a) and SAFMN-Large Sun et al. (2023), often introduce artifacts, such as blending the man's nose with the background or failing to reproduce textures. Similarly, diffusion-based methods, PASD Yang et al. (2024) and OSEDiff Wu et al. (2024a) exhibit excessive sharpness with content inaccuracies, omitting details like the man's mustache or failing to recover textures. In contrast, our LoRA-based models adeptly reconstruct high-fidelity details, particularly in complex areas with regular patterns, such as retaining the man's mustache and restoring textures to match the HR image. Overall, our LoRA-based approaches not only suppresses visual artifacts but also simultaneously restores structural shapes and realistic details.

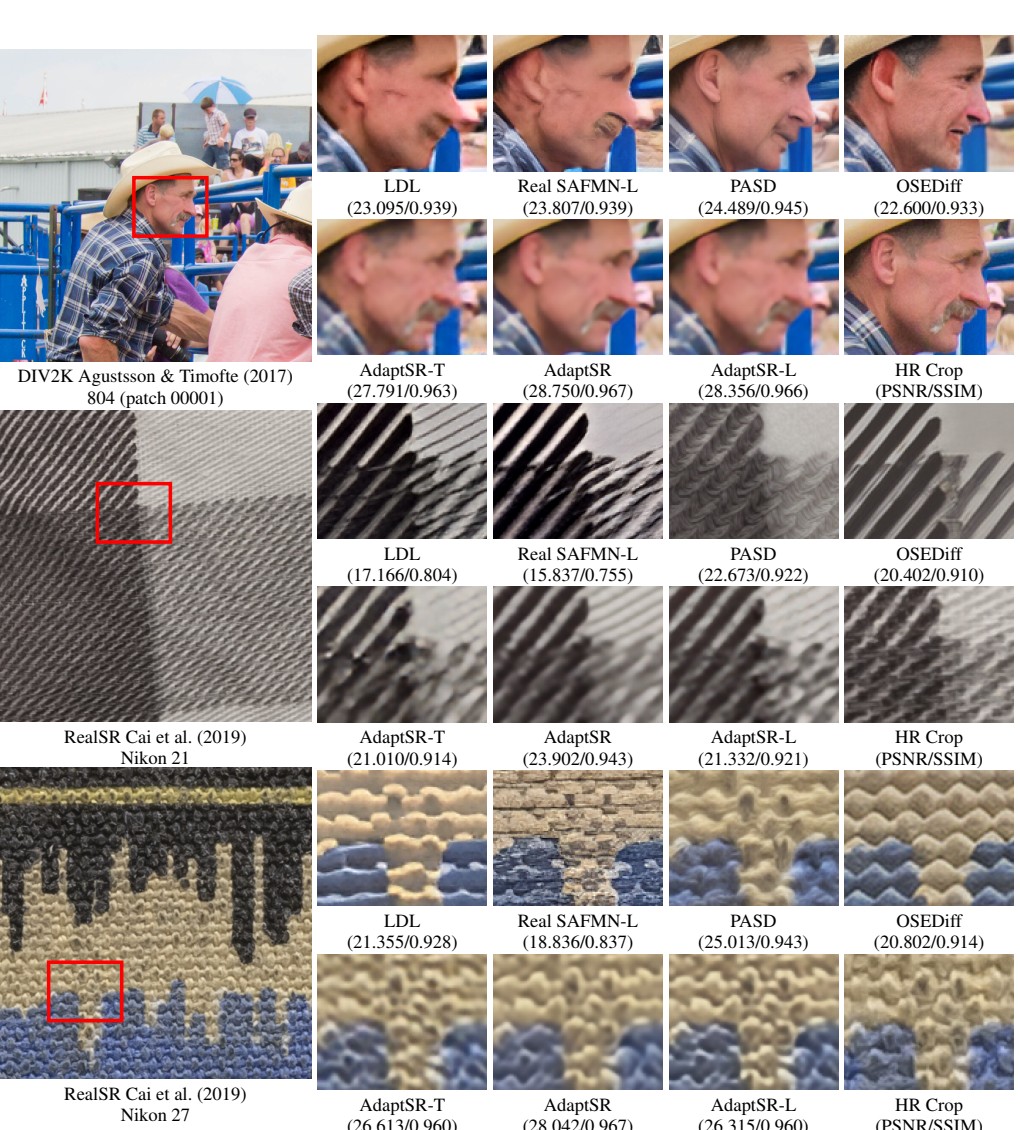

Figure 14: Visual comparison of the proposed LoRA-enhanced models (AdaptSR and its variants) with the state-of-the-art methods for ×4 real SR. GAN and diffusion models fail to capture the correct content of images, however, our AdaptSR models reconstruct high-fidelity and realistic details with correct alignment.

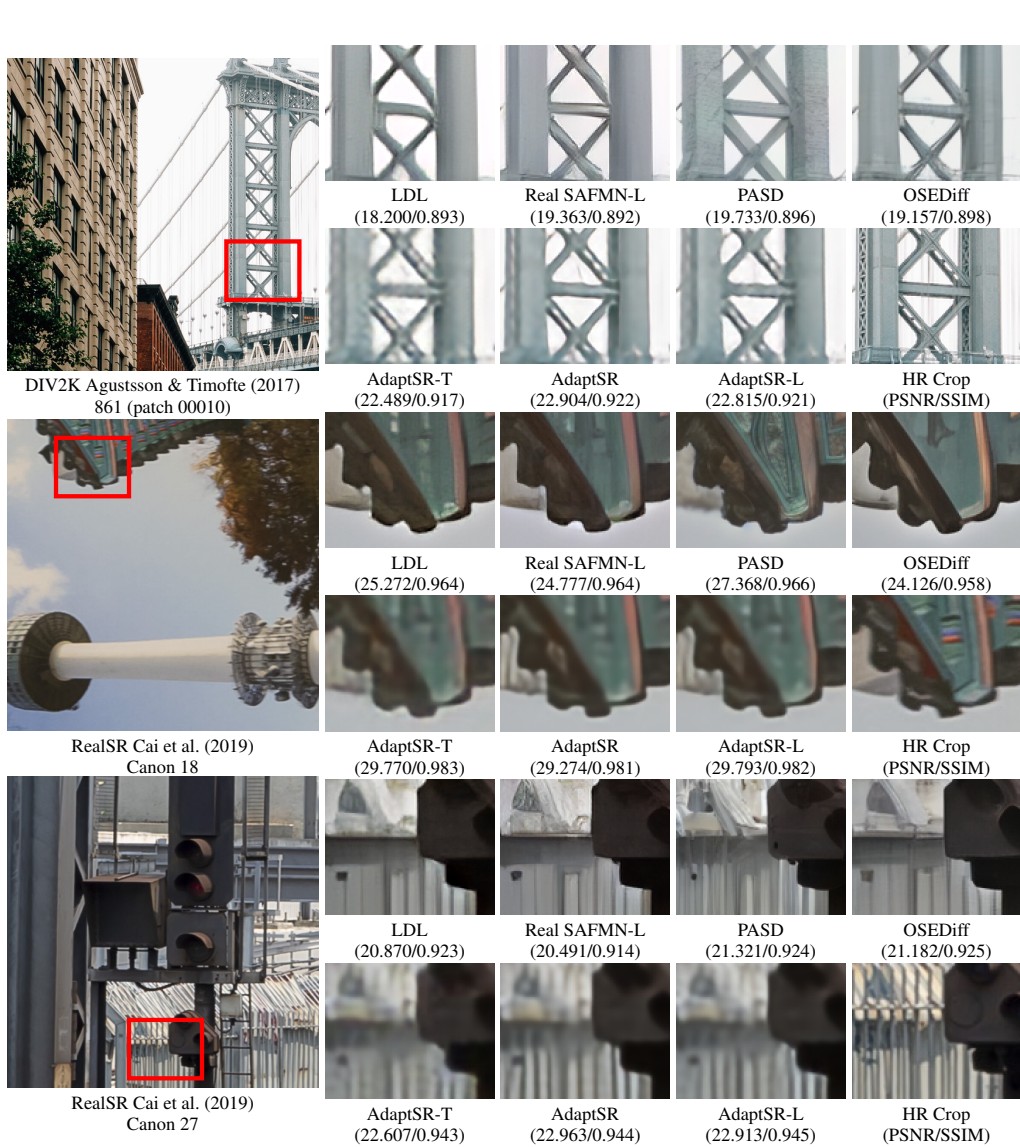

Figure 15: Visual comparison of the proposed LoRA-enhanced models (AdaptSR and its variants) with the state-of-the-art methods for ×4 real SR. GAN and diffusion models fail to capture the correct content of images, however, our AdaptSR models reconstruct high-fidelity and realistic details with correct alignment.