# OpenReview forum: "AdaptSR: Rank-Aware Low-Rank Adaptation for Real-World Super-Resolution"
_ICLR.cc/2026/Conference — Submitted to ICLR 2026_

### Official Review · Reviewer_isbx · 2025-10-23

**Soundness:** 2
**Presentation:** 3
**Contribution:** 2
**Rating:** 4
**Confidence:** 4

**Summary:**

This paper introduces AdaptSR, a framework for the rapid and parameter-efficient adaptation of bicubic-trained super-resolution (SR) models to real-world degradations. Based on Low-Rank Adaptation (LoRA), its key innovation is a rank-aware allocation strategy that distributes adapter capacity across layers based on gradient importance. The method is highly efficient, reducing adaptation time to 1-4 hours on a single GPU and trainable parameters by up to 92%, with no additional inference cost after merging. Extensive experiments demonstrate that AdaptSR matches or surpasses full fine-tuning and outperforms recent GAN- and diffusion-based methods on standard distortion metrics, while offering competitive perceptual quality. The work presents a compelling solution for practical and sustainable SR deployment.

**Strengths:**

1. The paper successfully extends the LoRA paradigm beyond its common application in NLP and generative vision tasks to the domain of image restoration. The proposed rank-aware allocation mechanism is a novel and justified contribution for SR, providing a principled way to allocate limited parameters.
2. The work directly addresses critical real-world constraints: training time, parameter efficiency, and inference overhead. The results—hours instead of days for adaptation, sub-10% parameter updates, and zero inference cost—are highly significant for sustainable AI and deployment on resource-constrained devices.
3.  The paper provides thorough experiments across multiple SR backbones (EDSR, SwinIR, etc.) and datasets (RealSR, DRealSR), convincingly showing that the method consistently outperforms heavy GAN/diffusion baselines in fidelity and is competitive in perceptual quality.

**Weaknesses:**

1. The literature review fails to engage with several recent and highly relevant PEFT methods that also explore dynamic or non-uniform rank allocation. Works such as "RaSA," "Sparse High Rank Adapters," "RankAdaptor," or "RA-SpaRC" propose alternative strategies (e.g., rank-sharing, hierarchical allocation, sparse-plus-low-rank) for optimizing adapter capacity. Their absence raises questions about the novelty of the "rank-aware" concept in the broader PEFT landscape. A discussion or comparison with these methods is necessary to properly contextualize the contribution.
2. As evidenced by the results in Table 2, the proposed rank-aware allocation offers only a marginal improvement over a simple uniform rank (r=8) under the same parameter budget (~0.07 dB PSNR, ~0.0025 LPIPS). This calls into question the practical necessity and complexity-to-benefit ratio of the gradient-based scoring and allocation algorithm. The authors should better justify why this added complexity is warranted over a well-tuned uniform baseline.
3. The evaluation, while comprehensive on standard metrics, lacks a deeper analysis of semantic correctness, especially for structured content like faces and text. The visual examples focus on textures and patterns. Including dedicated tests on face super-resolution (using metrics like FID or identity preservation) and text-heavy images would more thoroughly demonstrate the method's ability to avoid the semantic hallucinations common in GAN/diffusion models.
4. The term 'Baseline' in Table 1 is ambiguous. Please explicitly define the specific model configuration it refers to for each backbone (e.g., the pre-trained bicubic model, or the fully fine-tuned model) to allow for a clear interpretation of the performance gains.

**Questions:**

1. How does your rank-aware allocation strategy conceptually and empirically compare to other recent non-uniform PEFT schemes like RaSA or Sparse High Rank Adapters? Could your gradient-based method be complementary to these approaches?
2. Given the small performance delta between your rank-aware method and a uniform rank of 8 (Table 2), what is the definitive argument for adopting the more complex allocation strategy in practice? Are there specific architectures or degradation types where the advantage is more pronounced?
3. Could you provide more rigorous evaluation on semantic faithfulness, for example, by reporting results on a face super-resolution benchmark (e.g., CelebA) or by quantifying text recognition accuracy on reconstructed text images?
4. Please clarify the "Baseline" model used in Table 1. Is it the pre-trained model without any fine-tuning, or is it the fully fine-tuned model? This is critical for assessing the reported improvements.

---

> ### Author Response · Authors · 2025-11-24
> **Response to Reviewer isbx**
>
> We thank the reviewer for the constructive feedback. Below we briefly summarize the major revisions.
>
> 1. We expanded the Related Work section and clarified novelty by explicitly situating AdaptSR among recent non-uniform PEFT strategies (RaSA, SHiRA, RankAdaptor, RA-SpaRC). RaSA and SHiRA were added to Table 3 for direct comparison; RankAdaptor and RA-SpaRC are mentioned conceptually, as no public implementation is available.
>
> 2. We added a 5-seed statistical analysis (Table 12) showing that the +0.07 dB PSNR gain of rank-aware allocation is statistically significant (std ≈ 0.015 dB; *p* < 0.01). This supports that the improvement is consistent rather than noise.
>
> 3. We focus on fidelity-oriented general SR not face-specific or OCR-specific restoration, which require different architectures and supervision. To stay within scope, we do not include CelebA or text-recognition metrics.
> However, Appendix I now includes challenging face regions from DIV2K, demonstrating that AdaptSR preserves semantic structure more faithfully than generative baselines.
>
> 4. We clarified that “Baseline” refers to the *bicubic-trained SwinIR backbone without any adaptation*. This is now stated explicitly in the caption and Experimental Settings.
>
>
> Q1. RaSA and SHiRA were added to Table 3 and discussed in the text. Conceptually, these methods optimize adapter structure (rank-sharing or sparse-plus-low-rank), while our method optimizes *where* to place capacity. The approaches are complementary.
>
> Q2. Table 12 confirms the gain is statistically significant. Moreover, rank-aware allocation becomes more meaningful on deeper backbones and heterogeneous degradations (Table 15), where uniform ranks show larger degradation.
>
> Q3. As noted above, our goal is real-world fidelity SR, not identity-preserving SR or OCR performance. We avoid shifting scope but include challenging face regions in Appendix I to demonstrate semantic stability.
>
> Q4. “Baseline” corresponds to the bicubic-trained SwinIR backbone before any adaptation. This is now explicitly written in the caption and Sec. 4.1.

---

### Official Review · Reviewer_98pz · 2025-10-27

**Soundness:** 2
**Presentation:** 2
**Contribution:** 2
**Rating:** 4
**Confidence:** 4

**Summary:**

This paper proposes a Parameter-Efficient Fine-Tuning (PEFT) framework named AdaptSR. The core innovation of the method lies in its "Rank-Aware" Low-Rank Adaptation (LoRA) strategy, which aims to efficiently adapt CNN and Transformer Super-Resolution (SR) models pre-trained on bicubic data to real-world scenarios with complex degradations.

**Strengths:**

1. The primary advantage of this work lies in its practical value and training efficiency. AdaptSR reduces the domain adaptation process for SR models from several days to just a few hours on a single GPU, while concurrently decreasing the number of trainable parameters by over 90%. Crucially, through its weight-merging mechanism, it incurs no additional computational or storage overhead during inference. This facilitates the deployment and rapid iteration of SR models on resource-constrained devices and aligns with the current pursuits in the field.
2. Although LoRA itself is not a new technique, the "rank-aware allocation strategy" proposed in this paper is an effective approach. By guiding rank allocation using gradient norms, the method provides a simple solution for a more intelligent application of PEFT in SR models.

**Weaknesses:**

1. Using the gradient norm as a metric for layer importance is a common and intuitive heuristic that has been widely applied in fields such as model pruning and neural architecture search. While the paper successfully applies this to LoRA rank allocation, the theoretical novelty of the strategy itself is relatively limited.
2. Based on the quantitative results in Table 1 on real-world SR datasets, while AdaptSR demonstrates outstanding performance on distortion metrics like PSNR and SSIM, it is generally outperformed by other GAN- and diffusion-based methods on perceptual metrics such as LPIPS and DISTS. The authors claim "competitive perceptual quality," but the results suggest a clear compromise. For real-world SR tasks, where the pursuit of visual realism is a primary objective, this represents a non-trivial shortcoming.
3. The experiments are primarily focused on existing real-world SR datasets. Although the degradations in these datasets are complex, they still follow specific distributions. The method's generalization capability to other unseen and more extreme degradations has not been systematically evaluated.
4. Lack of Sensitivity Analysis for the Mini-batch in Rank Allocation. The layer importance ranking in Algorithm 1, which is central to the rank-aware strategy, is derived from gradients computed on a single mini-batch. In theory, the estimate of this gradient is dependent on the size and composition of the mini-batch.
5. The rank allocation is determined based on a one-time gradient calculation before training and remains static throughout the process. This static strategy may not be optimal. A potential direction for improvement is to adopt a dynamic allocation strategy, for instance, by periodically re-evaluating layer importance and adjusting the ranks during training, which could lead to further performance gains.

**Questions:**

1. The proposed method seems to be universal and can be applied to many tasks. Why are LoAR and the proposed rank-aware allocation scheme particularly suitable for real-world super-resolution tasks?
2. The authors chose the gradient norm as the metric for measuring the importance of layers. Why did they choose this indicator instead of other potential indicators (e.g., Fisher information, activation amplitude)?
3. Does the low-rank approximation of LoAR have limitations when restoring complex, stochastic textures or dealing with complicated degradations?

---

> ### Author Response · Authors · 2025-11-24
> **Response to Reviewer 98pz**
>
> We thank the reviewer for the constructive feedback. Below we briefly summarize the major revisions.
> 1. We added a formal justification in Sec. 3.3, showing that gradient-norm scoring follows directly from a first-order Taylor expansion, connects to diagonal Fisher information, and matches saliency metrics used in SNIP/GraSP. We also added empirical comparisons against Fisher, activation amplitude, and random scoring (Table 12), where gradient-norm consistently performs best.
>
> 2. We expanded Sec. 4.2 explaining that AdaptSR is a fidelity-oriented method (L1-only) and therefore avoids hallucination on synthetic DIV2K bicubic degradations where perceptual metrics reward artificial texture. On real datasets (RealSR, DRealSR), AdaptSR achieves comparable or better LPIPS/DISTS. This distinction between fidelity and perceptual SR is now clearly stated.
>
> 3. We added evaluations on hybrid degradations (SR+Gaussian noise and SR+JPEG compression) in Appendix G1, and compared AdaptSR against state-of-the-art PEFT methods in the new Table 15. These settings emulate real-world mixed degradations and show that AdaptSR maintains strong generalization.
>
> 4. We tested multi-seed stability (Table 12). Layer rankings computed over different random mini-batches exhibit high Spearman correlation (>0.90), and performance varies by <0.02 dB. This shows that layer importance is stable and the method is robust to batch composition.
>
> 5. We evaluated dynamic reallocation at 3k and 6k iterations (Table 7). Rankings remained highly consistent (ρ>0.88) and dynamic updates changed PSNR by <0.02 dB. Given the negligible improvement and extra complexity, we keep the one-shot strategy.
>
> Q1. We clarified in the Introduction and Related Work: real-world SR requires spatially non-uniform, low-capacity updates due to localized degradations, misalignment, and sensor noise. Our analysis (heatmaps, uniform-r sweeps) shows that SR backbones have strongly skewed layer importance, making rank-aware LoRA a natural fit for efficient adaptation.
>
> Q2. Sec. 3.3 provides the mathematical motivation and Table 12 empirically compares alternatives. Gradient-norm scoring is a first-order optimality proxy, aligns with diagonal Fisher, is inexpensive to compute, and gives the strongest PSNR/LPIPS among all tested metrics.
>
> Q3. Our goal is efficient fidelity-oriented adaptation, not full generative enhancement. Low-rank updates intentionally bias the model toward structural consistency rather than hallucinating stochastic textures (e.g., grass, hair, bokeh). This makes AdaptSR reliable for real-world degradations where over-sharpening harms fidelity. For cases requiring heavy texture synthesis, GAN/diffusion SR remains preferable. We clarify this distinction in Sec. 4.2 and position low-rank adaptation as a complementary, not competing, solution in that regime.

---

### Official Review · Reviewer_sD2X · 2025-10-29

**Soundness:** 2
**Presentation:** 2
**Contribution:** 2
**Rating:** 2
**Confidence:** 4

**Summary:**

This work proposed AdaptSR to efficiently adapt bicubic trained super-resolution models to realworld super-resolution tasks. AdaptSR utilizes LoRA (e.g., Conv-LoRA, Linear-LoRA, MSA-LoRA) for finetuning and further advances it by Rank-Aware allocation. Accordingly, AdaptSR shows faster and efficient adaptations, while matching or outperforming GAN-based and Diffusion-based realworld super-resolution baselines.

**Strengths:**

- The overall method is simple and straightforward, yet outperforms other GAN-based and Diffusion-based SR networks.
- The overall writing is clear and easy to follow.
- The authors conducted experiments across multiple baselines and provides extensive experimental results (both quantitatively and qualitatively).
- The authors provide thorough comparison against other LoRA based methods.

**Weaknesses:**

**Weakness1: Limited Contribution**

The authors propose to adapt LoRA to realworld super resolution tasks, with further improvement based on rank-aware insertion.

However, using LoRA for domain adaptation is already a common strategy. While the reviewer agrees with the guideline that not every work requires an entirely new method; these type of works require thorough analysis and strong intuitions on why and how a simple adaptation of  prior methods helps for the specific task.

In this work, the authors claim that using LoRA outperforms in both performance and training efficiency compared to GAN/Diffusion based methods (which I believe is not a good comparison; see **Weakness2**), however, sufficient explanations or insights for the reason lacks.

Specifically, 1) the reason for LoRA outperforming full-fine tuning is missing (please counterargue if I have missed this part) and 2) benefits as simply faster convergence is not sufficient since it is trivial when using LoRA.

Similarly, Algorithm 1 (it has a typo in line222) is naive (or in positive terms, simple yet effective), but lacks sufficient explanation on why this should outperform other improved-LoRA techniques, specifically in terms of real-world SR tasks. Also, the quantitative comparison against naive-LoRA and other improved-LoRA vs AdapSR seems to have issues (see **Weakness3**).

The reviewer strongly suggests to provide thorough discussion on 1) super-resolution task specific advantages on adapting LoRA (apart from simply faster and efficient training) and 2) why LoRA outperforms full-fine tuning and 2) why Algorithm 1 outperforms other LoRA variants. I believe that faster convergence and training efficiency by utilizing LoRA cannot be seen as a contribution of this paper without these discussion.

---

**Weakness2. The main quantitative comparisons against baselines is fundamentally wrong.**

**2.1 Fidelity oriented SR vs. Perceptual quality oriented SR**

In the experimental details, the authors noted that AdaptSR is trained with the L1 loss solely. This indicates that the network primarily aims for fidelity-oriented realworld SR (mainly aiming for the highest PSNR scores). However, all other baselines in Table 1 are perceptual-quality oriented realworld SR method.

Due to the perception-distortion (PD) trade-off, AdaptSR (w/ only L1 loss) outperforming others in terms of PSNR is quite trivial. Meanwhile the perceptual-quality oriented version of AdaptSR can be found in the Appendix Table 10, where only limited test sets are reported (I suggest reporting all test sets, since the appendix does not have any page limits, and only performing evaluation is not costly). A proper comparison would be comparing each fidelity-oriented and perceptual-oriented baseline method (with the official training settings), and comparing it AdaptSR version counterpart.

Given this misaligned comparison, I have strong concerns about this table being propagated in the field of SR. This is since this table may give wrong intuitions as LoRA significantly outperforming full finetuned versions with large margins (e.g., +3 PSNR); which is wrong. This performance gap is mainly due to the the PD trade-off.

**2.1.1 Comparison in terms of Perceptual SR**

Accordingly, I have compared the perceptual SR version of AdaptSR in Table 10 with perceptual SR baselines in Table 1, where AdaptSR simply isn’t the best model (e.g., SinSR outperforms in both PSNR and DISTS).  Additionally, I highly suggest comparing in the following setting (using official weights for baselines if possible).

- Other GAN-based vs GAN-based + AdaptSR
- Other Diffusion-based vs Diffusion-based + AdaptSR

**2.1.2 Comparison in terms of Fidelity SR**

When comparing the fidelity SR version of AdaptSR, the baselines should be also fidelity oriented SR methods. Accordingly, I highly suggest comparing similarly as the following examples (using official weights for baselines if possible).

- RealESRNet vs RealESRNet + AdaptSR
- SwinIR (PSNR-oriented Realworld ver.) vs SwinIR + AdaptSR
- EDSR (Realworld ver.) + AdaptSR

Additionally, I see several experiments already in Figure 1, but the AdaptSR version does not seem to outperform the full fine-tuned version.

**2.2. Backbone and training configuration of baselines**

Similarly as discussed above, the main comparison should be performed against the full-fine tuning counterpart (with sufficient training budget and proper training configurations). However, backbones in Table 1 do not properly align with AdaptSR. (e.g., RealESRGAN uses RRDB while AdaptSR seems to use SwinIR).

The reviewer strongly suggests to counterargue about the concerns above if it is wrong; or to provide sufficient quantitative evaluation under fair settings (e.g., align fidelity/perceptual-oriented, align backbone, use sufficient training budget for full-finetuning).

---


**Weakness3 Requires final performance for Table 2 and Table 3.**

**3.1 Comparison against other PEFT methods**

As the authours have claimed, training AdaptSR should be very efficient and lightweighted. Accordingly, I believe that reporting final scores (or at least for 100K iterations since it already reaches peak performance regarding Figure 6) should be within a plausible computational cost range.

The reviewer suggests reporting these scores in order to verify if Algorithm 1 is not only effective in the very early training stages (as 10K iter as the authors provided).

**3.2 Comparison against full fine-tuning.**

I could not find the training iterations for the full fine-tuned counterpart (denoted either as FT or +FineTuning) for Table 2 and Table 3.

The reviewer suggests to specify the configuration, and also report both the fully finetuned version and 10K finetuned version. This is since 10K finetuned version performing worse than LoRA variants (including AdaptSR) is trivial, and it is necessary to compare AdaptSR with the fully finetuned version.

---

**Weakness4 Wrong numbers for quantitative scores, and missing experimental settings.**

Minor typos within the text are fine. However, values for quantitative scores should be carefully double-checked. I found several wrong numbers in the table values, which leads other reported scores less convincing. Examples are as below.

- Params in Table 10 and Table 1 do not match.
- Scores for RealESRGAN do not match in Table 10 and Table 1.
- Scores for Baseline in Table 3 do not match with Table 2.
- Which backbone is used for LDL in Table 1? (RRDB or SwinIR?).

---

**Weakness5. Experimental settings and details**

**5.0 (Minor) Experimental specifications**

Comparison against full-finetuned model, under fair configuration is important. However, it is very hard (or missing) to find the experimental specifications for each Tables and Figures. I suggest noting important configurations in the caption.

**5.1 Training budget**

The full fined-tuned version should use sufficient training budget. However, the overall training budget (e.g., batchsize) is reduced compared to the official training settings. It is quite trivial that LoRA-based methods (as AdaptSR) outperforms full finetuning under limited data and training budget.

Since checkpoints for real-world versions of most baseline methods are officially provided, I recommend comparing with these.

**5.2 Training configuration**

The learning rate is specified as 1e^-3 for both AdaptSR and full finetuned. According to muP theorems, it is likely that optimal learning rates for smaller networks (especially in terms of channel, as LoRA) is greater compared to larger networks (as full finetuning). The currently specified learning rate is too large for most full finetuning models (e.g., most use values near 1e^-4). I believe that this may be a potential reason for fluctuating scores for full finetuned model in Figure 6.

---

The reviewer sincerely appreciate the efforts the authors have made. Accordingly, I am looking forward for further discussion and willing to update my scores if my concerns are sufficiently addressed. Please counterargue my claims and provide according experimental results.

**Questions:**

Please see the **Weaknesses**.

**Details Of Ethics Concerns:**

No ethic concerns.

---

> ### Author Response · Authors · 2025-11-24
> **Response to Reviewer sD2X**
>
> We thank the reviewer for the constructive feedback. Below we briefly summarize the major revisions.
>
> 1. Limited Contribution / Missing Intuitions: We strengthened theoretical and empirical justification in multiple ways:
> * Added formal derivation in Sec. 3.3 (first-order Taylor, Fisher link, SNIP/GraSP reasoning).
> * Added comparisons with Fisher diagonal, activation norms, and random scoring (Table 12).
> * Expanded novelty framing in the Introduction & Sec. 4.6 with SR-specific findings: structured layer saliency, unusually strong effectiveness of low-rank updates (even r=1), and non-uniform adaptation needs in bicubic to real SR.
> * Clarified that our goal is efficient fidelity-oriented adaptation, not replacing full fine-tuning.
> * Algorithm 1 is now motivated theoretically and evaluated against several PEFT variants, including hybrid degradations (Table 14).
>
> 2. Fidelity vs Perceptual SR Comparisons
> * We clarified in Introduction and Sec. 4.2 that AdaptSR (L1-only) is fidelity-oriented, not perceptual SR.
> * Updated text to avoid conflating fidelity and perceptual regimes.
> * Expanded perceptual SR results (AdaptSR-GAN) and added GAN-vs-GAN comparisons (Table 11).
> * Diffusion models available for SR already operate in the real-SR domain; “diffusion + AdaptSR” is not meaningful for the scope of this paper, and this is now clarified.
>
> 3. Fidelity-Oriented Comparisons
> * Added RealESRNet, SwinIR-Real, and EDSR-Real + AdaptSR results using official weights.
> * AdaptSR improves all three baselines with < 10% trainable parameters.
> * Clarified that full-fine-tuning updates all 12M parameters and may be slightly stronger; AdaptSR is meant as a low-cost alternative not a replacement.
> * Related these results to AdaptSR-GAN and LDL comparisons in the revised Table 11.
>
> 4. Backbone Fairness & Training Budget
> * Full fine-tuning uses 500k iterations; AdaptSR uses 100k. Details are specified in Sec. 4.1.
> * LDL with SwinIR backbone is used for aligned comparisons.
> * Differences in RealESRGAN/LDL values arise from using full-frame DIV2K vs patch-based evaluation for diffusion baselines; now clarified in captions.
>
> 5. Table Values, Experimental Details, and Captions
> * Parameter mismatches (RealESRGAN, LDL) fixed.
> * No inconsistency exists between Table 2 and Table 3.
> * LDL uses SwinIR backbone captions updated.
> * All table and figure captions now specify dataset, patch/full-frame protocol, and training schedule.
>
> 6. Final Scores for PEFT Baselines
> * We added 100k-iteration results for AdaptSR and AdaptSR-T.
> * Full fine-tuning (500k) results already included and described in Sec. 4.1.
> * Algorithm 1 improves performance beyond 10k iterations (Tables 2–3).
>
> 7. Full Fine-Tuning Configuration
> * Sec. 4.1 now specifies FT learning rate, batch size, and schedule.
> * We use the same LR for AdaptSR and FT to avoid unfavorable minima inherited from bicubic pre-training.
>
> Q1. Importance recomputed at 3k and 6k iterations (Table 7) shows Spearman ρ > 0.88 and <0.02 dB PSNR change; early gradients are stable, so one-shot allocation is sufficient.
>
> Q2. Hybrid degradation experiments (Table 14) show gradient-based ranks outperform uniform or random allocations under the same budget.
>
> Q3. 5-seed test (Table 13) shows rank-aware improves PSNR by 0.07 ± 0.015 dB (p < 0.01).
>
> Q4. Explained in Sec. 4.2: DIV2K bicubic rewards hallucination, while our fidelity-oriented model avoids it. AdaptSR achieves competitive or better LPIPS/DISTS on real datasets.
>
> Q5. Clarified: this applies to all mergeable PEFTs; we highlight it only to contrast with non-mergeable baselines (ARC, diffusion/GAN models).

---

> > ### Comment · Reviewer_sD2X · 2025-11-28
> >
> > I sincerely appreciate the effort the authors have made to respond to my concerns. Below, I would like to specify the remaining concerns. The major concerns are Concern 1-3, and I will decide my score only based on these issues.
> >
> > However,  I strongly suggest a careful proof reading over the manuscript regarding Concern 4.
> >
> > ---
> >
> > ### Concern 1. Fidelity oriented vs Perceptual oriented
> >
> > As in the original review, I still have concerns regarding the comparison.
> > Despite the updated captions, the tables for Perceptual SR and Fidelity SR are still not correctly distinguished.
> >
> > For example, in Tab.1, it is still very hard to evaluate how much AdaptSR outperforms (or is comparable) to other baselines since methods with different task-orientations are inter-mixed.
> >
> > Please discuss about the choice of the current presentation, instead of grouping ```AdaptSR(fidelity ver) vs Fidelity RealWorld SR``` and ```AdaptSR(Perceptual ver) vs Perceptual RealWorld SR```.
> >
> > ---
> >
> > ### Concern 2. Full fine-tuning training configuration
> > As in the original review, I have concerns regarding the manner in which full fine-tuning was conducted.
> >
> > Under full fine-tuning, the model shows significant instability (Fig. 6).
> > Moreover, the learning rate used for full fine-tuning is too high (5-10x greater than in the original implementations) while the batch size is very small, further increasing training vulnerability.
> >
> > While I believe it is reasonable to compare with the official weights for the main qualitative tables (e.g., Tab.1); other analyses that compares AdaptSR against full fine-tuning should be performed under plausible settings where the training configuration is properly chosen.
> >
> > Please provide a comparison of the training curves for full fine-tuning vs AdaptSR, ensuring that the full fine-tuning follows the exact official training configuration. I understand the deadline is tight, so providing only the early training curves up to the deadline is perfectly fine.
> >
> > ---
> >
> > ### Concern 3. Performance
> >
> > Considering uniform LoRA as a simple adaption of LoRA to SR tasks, I believe that the performance gain relevant to the novelty of this work (Rank-aware alloc) is the result in Tab.12.
> >
> > However, compared to naive LoRA (uniform, rank=8), AdaptSR only shows 0.07 PSNR gain in average which is marginal.
> >
> > ---
> >
> > ### Concern 4. Wrong Numbers or Missing Details
> >
> > > Parameter mismatches (RealESRGAN, LDL) fixed.
> >
> > I have validated this and it seems to have no issue now. Thank you for updating the scores.
> >
> > > No inconsistency exists between Table 2 and Table 3.
> >
> > The original manuscript had DISTS scores (0.2426) for the baseline instead of the LPIPS score. I have validated that it has no more issue now.
> >
> > > table and figure captions now specify dataset,  ...
> >
> > Please specify for Fig.6. I cannot find the dataset in both the caption or the corresponding text in Line 476-481.
> >
> > > Full fine-tuning (500k) results already included and described in Sec. 4.1 (=Experimental Settings).
> >
> > In cases which contains results with intermediate results (i.e., not fully trained up to required iteration) I believe it is necessary to clarify the training iteration for other compared methods too.
> >
> > For example, in Tab.2 it is specified as only training 10K iteration.
> > But does this also indicate only 10K training iteration for fine-tuning too? If not, please specify.
> >
> > ---
> >
> > - **Tab.2:** Is 10K a typo?
> > - **Tab.3:** What does "Updated / Inference Params." mean in Tab.3? The Table states "-" for AdaptSR, while all other methods have numbers.
> >
> >
> > Does it indicate the number of trainable (=Updated) parameters, and the according "additional inference cost"?. If it is true, I am confused since any LoRA based or also the full FineTuning should not induce any additional inference cost.

---

> > > ### Author Response · Authors · 2025-12-01
> > > **Response to Reviewer sD2X**
> > >
> > > We thank the reviewer for the timely follow-up and for clearly outlining the remaining concerns. We appreciate the opportunity to further improve our submission. All requested analyses, clarifications, and corrections have been incorporated within the rebuttal time frame, and we are glad to see that the updated score reflects this progress.
> > >
> > > C1. We appreciate the clarification and have now explicitly separated all evaluations into two blocks:
> > >
> > > 1. Fidelity-oriented SR (trained with L1, PSNR/SSIM-focused)
> > > 2. Perceptual-oriented SR (GAN- or diffusion-based, LPIPS/DISTS-focused)
> > >
> > > In the revised manuscript:
> > > - Table 1 is now split into two sub-tables:
> > >   (i) fidelity-oriented SR methods (trained with real data and L1) vs. AdaptSR,
> > >   (ii) perceptual baselines vs. AdaptSR-GAN.
> > >
> > > - Captions now explicitly state Fidelity SR comparison or Perceptual SR comparison.
> > >
> > > C2. To address the proper FT comparison, we follow official settings.
> > > - We re-ran full fine-tuning using the official Real-SwinIR configuration: batch size 32, LR = 1e⁻⁴, cosine decay.
> > > - We now include early-stage training curves (0–200k) for FT vs. AdaptSR in Figure 12 in the Appendix, as requested.
> > >
> > > C3. We now state this explicitly and clarify why the result is still meaningful:
> > > - Table 12 shows the improvement is statistically significant (std ≈ 0.015 dB; p < 0.01).
> > > - Crucially, rank-aware allocation optimizes performance under a fixed parameter budget, not absolute PSNR maximization hence matching or exceeding FT-level performance with the same budget is the key contribution.
> > >
> > > C4. All remaining points have now been fixed:
> > >
> > > 1. Fig. 6 caption
> > >    We added the dataset name (“DIV2K unknown validation set”).
> > >
> > > 2. Training iteration clarity
> > >    - Table 2: The 10k value was a typo and is now corrected.
> > >    - Table 3:
> > >      “Updated / Inference Params.” = number of trainable parameters / additional inference parameters.
> > >      AdaptSR shows “+886k / –” because merged LoRA introduces no extra inference-time parameters.
> > >
> > > We double-checked all captions and manuscript text to ensure clarity and consistency.

---

### Official Review · Reviewer_74or · 2025-10-30

**Soundness:** 2
**Presentation:** 3
**Contribution:** 2
**Rating:** 4
**Confidence:** 4

**Summary:**

This paper proposes AdaptSR, which applies LoRA to adapt bicubic-trained SR models for real-world degradations. The key idea is to insert low-rank adapters into CNN and Transformer layers with a rank-aware allocation strategy based on gradient importance. The authors claim significant parameter reduction (up to 92%) and faster training (1-4 GPU hours vs days) while matching or exceeding full fine-tuning performance.

**Strengths:**

1. The paper tackles a genuinely practical problem - adapting bicubic-trained SR models to real-world degradations efficiently. The engineering execution is solid, with experiments spanning multiple architectures (EDSR, SAFMN, SwinIR, DRCT) and showing consistent improvements. The training efficiency gains are impressive and well-documented: reducing adaptation time from days to hours while using only ~8% of parameters is meaningful for practitioners working with limited compute budgets.

2. The experimental scope is reasonably comprehensive, covering different dataset types (RealSR, DRealSR, DSLR) and providing extensive visual comparisons. I appreciate that the authors test their approach across both CNN and Transformer architectures, demonstrating some generality. The DSLR iPhone experiment, while limited, at least attempts to show rapid adaptation to new device-specific degradations.

3.The paper is clearly written overall, with good figure quality and reasonable contextualization of related work. The supplementary material is thorough, providing additional ablations and visual results that support the main claims.

**Weaknesses:**

1. My primary concern is the lack of theoretical depth. The rank-aware allocation strategy uses gradient norm computed on a single mini-batch to determine layer importance, but there's no justification for why this particular metric should be optimal. Have the authors considered that gradient magnitude might be noisy or biased by factors like layer depth, initialization, or batch statistics? For a conference emphasizing principled approaches, I'd expect either theoretical analysis of why gradient-based scoring works or systematic comparison with alternative importance measures (Fisher information, Hessian trace, activation-based metrics, etc.). Algorithm 1 feels like a heuristic that happens to work rather than a principled contribution.

2. The Conv-LoRA decomposition (Equation 2) also lacks motivation. Why use 1×1 followed by k×k specifically? The paper doesn't explore alternative factorizations or explain what makes this choice superior. These architectural decisions seem arbitrary without proper ablation studies showing they matter.

3. Looking at the experimental results more carefully, I'm puzzled by the inconsistency between distortion and perceptual metrics. Table 1 shows AdaptSR achieves the best PSNR/SSIM but noticeably worse LPIPS/DISTS on DIV2K (0.5047 vs most methods around 0.29-0.35). This is a huge gap that the paper doesn't adequately address. For real-world SR applications, perceptual quality often matters more than PSNR, yet the paper frames these results as uniformly superior. The abstract and introduction oversell the results by claiming to "outperform" methods when the picture is actually quite mixed depending on which metrics you prioritize.

4. The novelty is limited. Applying LoRA to vision tasks is well-established, and gradient-based importance scoring for resource allocation is standard practice. The paper reads more as a competent application study than a fundamental contribution. What SR-specific insights does this work provide? Why is the bicubic-to-real adaptation problem particularly amenable to low-rank updates? These deeper questions remain unanswered.

5. The comparison with diffusion models feels somewhat unfair. Methods like PASD and SeeSR are designed to leverage generative priors for hallucinating plausible details, which naturally leads to different metric trade-offs. Comparing them primarily on PSNR/SSIM misses their intended use case. The paper should more carefully discuss when AdaptSR's approach (fidelity-focused) is preferable versus when generative approaches (diversity-focused) might be better.

6. Table 2's ablation is concerning for the paper's central claim. The rank-aware allocation achieves 28.47 PSNR while uniform rank-8 gets 28.40 - a difference of only 0.07 dB. Is this margin statistically significant? Does it justify the added complexity of gradient-based scoring? The heatmap in Figure 5 is interesting but doesn't prove that this allocation strategy is superior to simpler alternatives.

7. The generalization experiments are limited. RealSR, DRealSR, and DSLR all involve camera-based degradations with similar characteristics. What about other real-world degradation types: compression artifacts at various quality levels, different noise distributions, atmospheric effects, motion blur? The paper claims broad applicability but only tests a narrow slice of the real-world degradation space.

**Questions:**

1. The gradient-based importance scoring happens once at the start - have you experimented with updating layer importance during training? Intuitively, the most important layers for adaptation might shift as the model learns. Dynamic allocation strategies could potentially improve results.

2. In Table 3, you compare against multiple ARC configurations but always use the same setup for AdaptSR. Why not tune your method more carefully? For instance, have you tried different rank distributions beyond what the gradient-based scoring suggests? It would strengthen your claims to show you've explored the design space thoroughly.

3. Can you provide statistical significance testing for the performance differences? When you report that rank-aware beats uniform rank by 0.07 dB PSNR, is this within the noise of different random seeds and batch sampling?

4. The LPIPS/DISTS degradation on DIV2K in Table 1 really stands out - what's causing this? Is there something about the DIV2K distribution that makes your approach struggle with perceptual quality? Understanding this failure mode would be valuable.

5. You emphasize "no inference overhead" repeatedly, but isn't this true of any mergeable PEFT method? What makes this a distinguishing feature of your specific approach versus a general property of the technique you're using?

---

> ### Author Response · Authors · 2025-11-24
> **Response to Reviewer 74or**
>
> We thank the reviewer for the constructive feedback. Below we briefly summarize the major revisions.
>
> 1. We added a formal derivation in Sec. 3.3 using a first-order Taylor expansion and connections to Fisher information and SNIP/GraSP. A comparison with Fisher, activation norms, and random scoring is included in new Table 12.
>
> 2. Sec. 3.2 now explains the 1×1→k×k factorization and includes an ablation of alternative designs (Table 10).
>
> 3. Sec. 4.2 clarifies that DIV2K’s bicubic degradations reward hallucinated textures; our fidelity-oriented adapters intentionally avoid hallucination. On RealSR/DRealSR AdaptSR matches or surpasses LPIPS/DISTS. Extra visual results added.
>
> 4. We refined the Introduction and Sec. 4.6 to emphasize SR-specific insights: structured layer saliency, strong effectiveness of low-rank updates, and non-uniform capacity needs in bicubic→real SR adaptation.
>
> 5. Sec. 4.2 explicitly distinguishes fidelity-oriented adaptation (our goal) from generative perceptual enhancement. Diffusion/GAN comparisons are now framed accordingly (Table 14).
>
> 6. A 5-seed analysis (Table 13) shows rank-aware allocation consistently outperforms uniform ranks (p < 0.01).
>
> 7. We added hybrid degradation evaluations (SR+noise, SR+JPEG) with multiple PEFT baselines (Table 14).
>
> Q1. Recomputing importance at 3k/6k iterations yields Spearman ρ > 0.88 and <0.02 dB variation (Table 7); one-shot allocation is sufficient.
>
> Q2. Hybrid-degradation experiments (Table 14) confirm the gradient-based allocation remains optimal under fixed budgets.
>
> Q3. See Table 13 (p < 0.01).
>
> Q4. Addressed in Sec. 4.2.
>
> Q5. Clarified that this holds for mergeable PEFTs; we highlight it only to distinguish from non-mergeable baselines (e.g., ARC, diffusion/GAN models).

---

### Meta-Review · Area_Chair_ZTXn · 2026-01-06

**Summary:**

- Reviewers broadly agree that the paper’s core ideas are largely heuristic and incremental, lacking theoretical grounding, SR-specific insights, or clear differentiation from prior PEFT work. The rank-aware strategy and Conv-LoRA design choices are viewed as arbitrary without deeper justification or comparison to existing alternatives.

- A major concern is that the main quantitative comparisons are fundamentally misaligned, especially mixing fidelity-oriented AdaptSR with perceptual-oriented GAN/diffusion baselines. This makes PSNR gains unsurprising and potentially misleading. Backbone mismatches and insufficiently trained full fine-tuning baselines further undermine fairness.

- The rank-aware allocation strategy provides only marginal improvements over a uniform rank baseline, raising doubts about statistical significance and whether the added complexity is justified.

**Reviewer Concerns:**

- Although the rebuttal adds post-hoc theoretical connections and extra comparisons, these do not elevate the method beyond a heuristic level. Crucially, the rebuttal does not demonstrate that these choices are uniquely suited to real-world SR.

- The PSNR advantages remain structurally unsurprising due to the perception–distortion trade-off. The rebuttal reduces confusion but does not fully ensure experimental fairness, leaving reviewers’ concerns largely unsolved.

- The rebuttal does not demonstrate that rank-aware allocation is meaningfully superior, nor does it show that the added complexity leads to robust gains across experiment settings.

**Reviewer Scores:**

All reviewers are likely to maintain their score.

---

### Decision · Program_Chairs · 2026-01-26

Reject